# Phenotypic heterogeneity follows a growth-viability tradeoff in response to amino acid identity

Kiyan Shabestary [1], Cinzia Klemm[1], Benedict Carling [1,2], James Marshall [1,2], Juline Savigny[1], Marko Storch [2,3] & Rodrigo Ledesma-Amaro [1]

In their natural environments, microorganisms mainly operate at suboptimal growth conditions with fluctuations in nutrient abundance. The resulting cellular adaptation is subject to conflicting tasks: growth or survival maximisation. Here, we study this adaptation by systematically measuring the impact of a nitrogen downshift to 24 nitrogen sources on cellular metabolism at the single-cell level. *Saccharomyces* lineages grown in rich media and exposed to a nitrogen downshift gradually differentiate to form two sub-populations of different cell sizes where one favours growth while the other favours viability with an extended chronological lifespan. This differentiation is asymmetrical with daughter cells representing the new differentiated state with increased viability. We characterise the metabolic response of the sub-populations using RNA sequencing, metabolic biosensors and a transcription factor-tagged GFP library coupled to high-throughput microscopy, imaging more than 800,000 cells. We find that the subpopulation with increased viability is associated with a dormant quiescent state displaying differences in MAPK signalling. Depending on the identity of the nitrogen source present, differentiation into the quiescent state can be actively maintained, attenuated, or aborted. These results establish amino acids as important signalling molecules for the formation of genetically identical subpopulations, involved in chronological lifespan and growth rate determination.

Most microorganisms spend most of their lifetime in a non-growing, quiescent state[1–3]. Upon occasional exposure to nutrients, they exit this state and resume growth. In microbes such as yeast, quiescence and proliferative growth are fundamentally opposite cellular states with very distinct gene expression profiles and metabolic signatures[4–8]. While the metabolism of growing cells is dominated by anabolic reactions, quiescent cells rely on catabolism for survival and typically undergo important metabolic rewiring associated with an upregulation of the stress response, recycling of internal macromolecules and an overall reduced metabolic activity[4,6,9,10]. Physiologically, quiescent cells are smaller and possess a thicker cell wall that provides resistance to a wide variety of stresses[4] and different quiescence states can be accessed depending on the environmental insult experienced[9].

Understanding how microorganisms regulate their cell size, growth rate and survivability in response to environmental signals including starvation has been a major challenge in quantitative cellular physiology[11–14]. Studies at the population level have drawn empirical relationships between cell growth, cell size and nutrient availability[15–17].

[1]Department of Bioengineering and Imperial College Centre for Synthetic Biology, Imperial College London, London SW7 2AZ, UK. [2]London Biofoundry, Imperial College Translation & Innovation Hub, London, UK. [3]Department of Infectious Disease, Imperial College London, London, SW7 2AZ, UK. ✉e-mail: k.shabestary@imperial.ac.uk; r.ledesma-amaro@imperial.ac.uk

Yet, population-averaged observations often mask single-cell behaviours due to phenotypic variations across cells[14,18,19]. In particular, cell-to-cell heterogeneity is often found in a population of genetically identical (isogenic) cells, even when growing under steady-state assumptions, due to differences in stochasticity, cell ageing or cell cycle progression[14,20–24]. Recent advances in single-cell phenotyping such as cell segmentation and tracking in microscopy[25–27] or single-cell RNA sequencing[22,23,28–31] have given new insights into the emergence of phenotypic heterogeneity in microorganisms. While the underlying differentiation processes are still poorly understood, the fitness benefits are clear. Phenotypic heterogeneity, or population multimodality[32], denoting the presence of two or more distinct isogenic subpopulations (Fig. 1a), can reduce the risks of investing all resources towards a specific phenotype as in bet-hedging[20,33–35] or can lead to a more efficient proteome partitioning through cellular differentiation or division of labour as in multicellular organisms[36–38].

Here, we study the single-cell response to suboptimal growth conditions in the model eukaryote *Saccharomyces cerevisiae*. Using nitrogen downshift as a case study, we report the presence of both isogenic quiescent and growing subpopulations displaying differences in cell size, chronological lifespan and growth resumption capability. Based on previously published single-cell RNA sequencing datasets[23], we identify subpopulation markers that allow high-throughput interrogation of cellular fate at the onset of a nitrogen downshift and study differentiation across 24 different nitrogen sources present in limited or replete amounts (Fig. 1b). We perform a multi-omics analysis of the differentiation process using subpopulation RNA sequencing and analyse the single-cell metabolic response using a prototrophic GFP-tagged transcription factor library and metabolic biosensors (Fig. 1a, Supplementary Fig. 1). Our results reveal the presence of two distinct subpopulations reflecting a global population-wide strategy where isogenic subpopulations are metabolically specialised in either growth

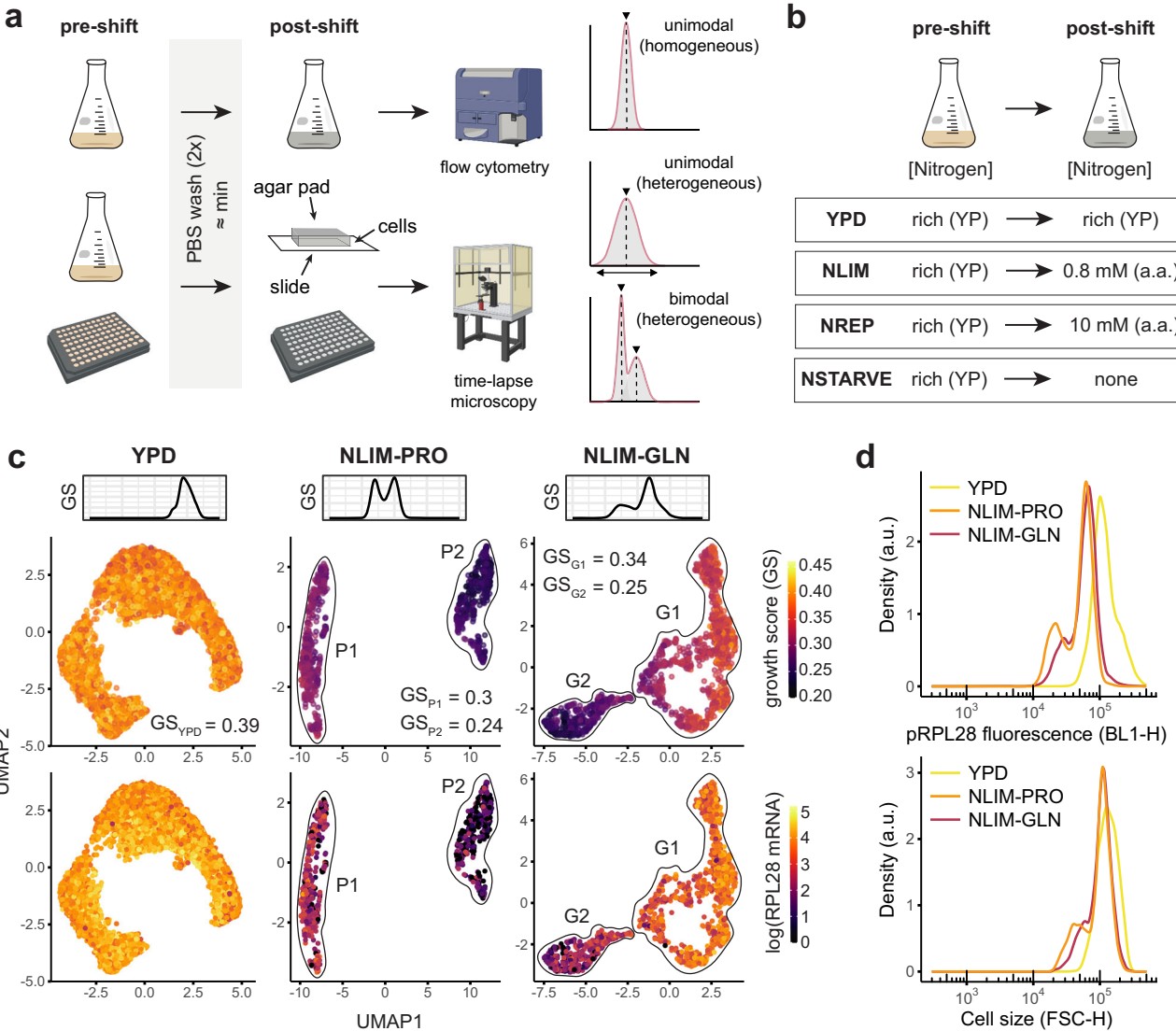

**Fig. 1 | Nitrogen downshifts lead to phenotypic heterogeneity. a** Conditions tested throughout this study. Cells were grown in rich (YPD) media until they reached exponential phase, washed twice with PBS and resuspended in one of the media. Different modalities (unimodal or bimodal) were observed across conditions. Flow cytometer and microscope schematics were made with BioRender.com released under a CC-BY-NC-ND 4.0 International license. **b** Pipeline used throughout this study to monitor single-cell differentiation. **c** scRNAseq datasets were obtained from ref. 23 and describe cells shifted to 0.8 mM proline (NLIM-PRO) or glutamine (NLIM-GLN) as well as the control (YPD). Plots represent dimensionality-reduced data using UMAP, where each point represents a single cell. Cells in the top UMAP plot are coloured by growth scores, calculated from a regression model[39] trained on bulk RNAseq data. Histogram above represents the density of growth scores (GS) for each condition. **d** Flow cytometry data for cells exposed to a nitrogen downshift. GFP fluorescence (BL1-H channel) and cell size (FSC-H forward scatter channel) were used to measure single-cell heterogeneity. Arbitrary units are shown (abbreviated a.u.).

or viability depending on the nitrogen source present. Results obtained in wild and laboratory *Saccharomyces* strains suggest a previously unknown amino acid-dependent and conserved behaviour shaping population dynamics.

## Nitrogen shift leads to phenotypic heterogeneity

To study population adaptation following a nitrogen shift, we took advantage of recently available yeast single-cell RNA sequencing datasets (scRNAseq). Jackson et al.[23] observed that diploid prototrophic *S. cerevisiae* cells, generated from FY4 and FY5 laboratory strains, exposed to nitrogen downshifts display two subpopulation clusters with distinct transcriptome profiles. The differentiation process was also impacted by the quality of the nitrogen source where growth on the non-preferred amino acid proline gave a stronger differentiation than for the preferred amino acid glutamine (Fig. 1c). We further leveraged this data to find subpopulation markers that could be used to study the emergence of this differentiation. Remarkably, in both conditions, one of the subpopulation clusters (cluster P2 for proline and G2 for glutamine) showed a decrease in expression of ribosomal genes while having a higher stress signature (Fig. 1c). We investigated whether this difference in transcriptome could translate into a difference in growth. Using a previous regression model predicting growth rate based on bulk transcriptomic data obtained from nutrient-limited chemostats[39], we found that these clusters (P1/P2 and G1/G2 for proline and glutamine, respectively) had distinct predicted growth profiles 4 h into nitrogen limitation according to scRNAseq data (Fig. 1c). We identified a subpopulation marker RPL28 (YGL103W) whose transcript levels were high and significantly different between G1 and G2, as well as between P1 and P2 (Supplementary Fig. 2 and Fig. 1c). Flow cytometry of isogenic BY4741 laboratory cells containing super-folded GFP (sfGFP) under the control of pRPL28 promoter (700 bp upstream of start codon) was first used to reproduce the observed heterogeneity at 4 h post-shift. As suggested from scRNAseq data, bimodality could be detected in nitrogen-limited media with proline or glutamine as nitrogen source but not in rich Yeast extract-Peptone-Dextrose (YPD) media (Fig. 1d). For each of these nitrogen-limited conditions, two subpopulations of cells could be detected for both pRPL28-sfGFP intensity and cell size (Fig. 1d). Scatter plot of pRPL28 fluorescence versus cell size shows that the "low" subpopulation with the lower pRPL28 fluorescence had smaller cell size (lower FSC forward scatter signal), while the "high" subpopulation had a cell size and a pRPL28 fluorescence that were closer to pre-shift levels in YPD (Fig. 1d, Supplementary Fig. 3a). To confirm that these differences in pRPL28 fluorescence between low and high subpopulations were in fact not due to cell size differences, we further normalised fluorescence by cell size (Supplementary Fig. 3b). When the subpopulations were clustered based on cell size and cell size-normalised pRPL28 fluorescence, cell size-normalised pRPL28 fluorescence was still significantly different ($p < 0.005$; two-sided unpaired $t$ test) between subpopulations for both proline and glutamine conditions, indicating variations in cell size alone could not account for the differences observed in pRPL28 fluorescence (Supplementary Fig. 3c, d). Additionally, exposing Rpl28 tagged GFP strains to 4 h of proline treatment could also show the emergence of a "low" subpopulation with lower Rpl28-GFP levels (Supplementary Fig. 3e), indicating that lower fluorescence in the "low" subpopulation is not due to post-transcriptional regulation of sfGFP alone.

To further investigate whether high and low subpopulations were truly isogenic, we sorted cells based on forward side scatter and GFP fluorescence, separated the two subpopulations, and performed the same shift from YPD to nitrogen limitation for high and low fractions separately. Given the significant overlap in cell size and pRPL28 intensity between both subpopulations at the onset of the shift (Fig. 2a), sorting on both fluorescence and cell size allowed for a more precise separation at the early stage of differentiation (Supplementary

Note 1). Regardless of their post-sorting classification into low or high subpopulations, both subpopulations re-exposed to rich media could regenerate both low and high fractions when shifted again, indicating that the heterogeneity observed was phenotypic and reversible when re-exposed to rich media (Supplementary Note 1). Extending the washing step and leaving the cells in PBS for 2 h did not affect the dynamics of heterogeneity (Supplementary Fig. 4).

## Low subpopulation is a daughter-specific reversible quiescent state

To investigate the dynamics of the differentiation between high and low subpopulations, we performed a time-course analysis for nitrogen-limited proline and glutamine conditions and monitored cell size and pRPL28 fluorescence every 2 h for 8 h (Fig. 2a). The emergence of a second subpopulation could be seen for both conditions after 2 h and was more pronounced after 4 h. After that, we noticed a diverging outcome between conditions where cells in the nitrogen-limited proline (hereafter NLIM-PRO) condition maintained bimodality while cells in the glutamine conditions (NLIM-GLN) became more unimodal, but yet heterogeneous, in cell size and pRPL28 fluorescence over time (Fig. 2b, Supplementary Fig. 5a, b). Bimodality also led to significant differences in the cell size coefficient of variation (CV), defined as the population cell size standard deviation divided by its mean, always higher for NLIM-PRO throughout the differentiation (Supplementary Fig. 6). To further evaluate the number of cells in each subpopulation, we clustered cells based on cell size and pRPL28 fluorescence (see Methods) and assumed two distinct subpopulations based on previous histograms (Fig. 1d). Given the difficulty to reliably cluster subpopulation early and late into the shift, we assigned subpopulations based on the clustering performed on 4 h NLIM-PRO, which gave the clearest clustering (Supplementary Fig. 7 and Fig. 2b). Significantly more cells were assigned to the low subpopulation for NLIM-PRO compared to NLIM-GLN from 4 h onwards (Fig. 2c).

The strong and weak bimodality observed for proline and glutamine, respectively, echoed the growth score distributions computed from scRNAseq (Fig. 1c). To confirm that the observed subpopulations were indeed the scRNAseq clusters and analyse the early response at the same time, we performed RNA sequencing on the sorted fractions separately (subpopRNAseq) 1 h post-shift (Supplementary Data 1, Supplementary Data 2, Fig. 2d). Comparing genes that were significantly different ($p_{adj} < 0.05$) between subpopulations and clusters in the NLIM-PRO condition revealed a good correlation (Spearman coefficient $\rho = 0.66$) between the high subpopulation and cluster P1 as well as between the low subpopulation and cluster P2 despite differences in experimental set-up, laboratory strain and sampling time (1 h for subpopRNAseq vs 4 h scRNAseq) (Supplementary Fig. 8). SubpopRNAseq confirmed the upregulation of G1-phase daughter-specific markers DSE1 and DSE2 for the low subpopulation as seen in cluster P2 in the scRNAseq dataset (Fig. 2d, Supplementary Fig. 8). Genes involved in cytokinesis (HOF1, BUD4, ACE2, CHS2) or chromosome segregation (HCM1) were significantly ($p_{adj} < 0.05$, Benjamini−Hochberg procedure) up-regulated in the high subpopulation as well as the mitotic exit regulators SPO12, DBF2 and CDC5 (Supplementary Data 2). KEGG enrichment of differentially expressed genes between high and low proline subpopulations revealed major differences in cell cycle progression, meiosis, MAPK signalling pathway and DNA repair (Supplementary Fig. 9). Further analysis of scRNAseq clusters 4 h into the shift shows that the low subpopulation had all the hallmarks of quiescence with a reversible growth arrest and upregulation of quiescence markers[4] as well as proteins involved in autophagy (Supplementary Note 2, Supplementary Data 1).

Enrichment of the daughter-specific markers DSE1 and DSE2 in the low quiescent subpopulation prompted us to investigate subpopulation dynamics further. First, we sorted high and low subpopulations grown for 2 h in NLIM-PRO or NLIM-GLN and tracked their evolution

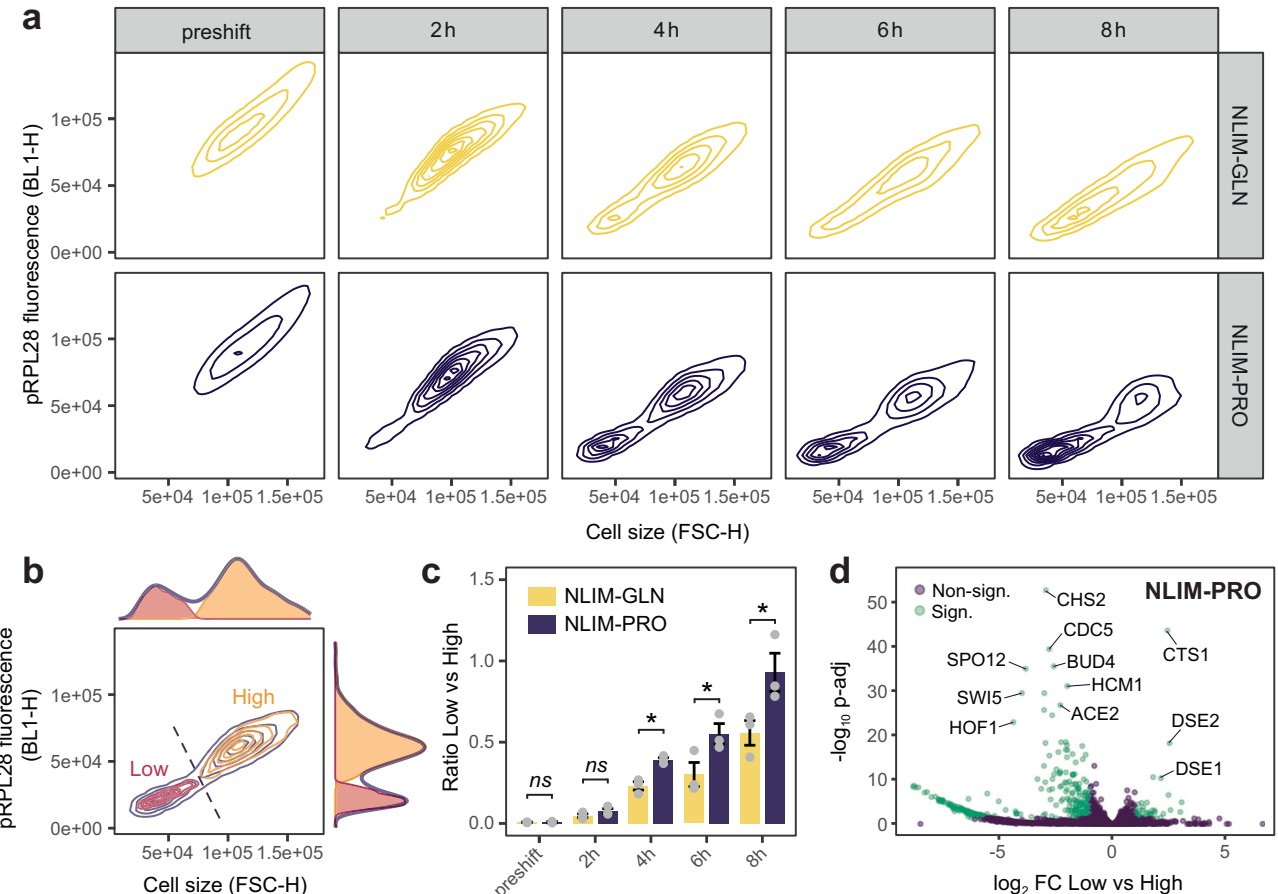

**Fig. 2 | Dynamics of cellular differentiation in NLIM media. a** Contour plots of cell size (FSC-H) and pRPL28 fluorescence (BL1-H) over time for NLIM-PRO (dark purple) and NLIM-GLN (yellow). Pre-shift indicates growth in YPD prior to the shift. Contours indicate areas of higher density. **b** Calculation of subpopulation fractions for each timepoint and condition based on clustering applied to **a**. Example shown is NLIM-PRO after 4 h where two multivariate Gaussians were fitted to the data (Expectation Maximisation; Methods). Clustering was performed on cell size and pRPL28 fluorescence (contour plot) and is shown for both dimensions separately (histograms). Light orange and light red colours represent high and low fractions, respectively. **c** Based on this clustering, hard assignment (dashed line) was generalised to other conditions and timepoints to calculate subpopulation ratios (bar chart) of low versus high. Bar plot represents the mean of biologically independent samples ±SEM ($n = 3$) taken on different days. Significance scores denote the $p$ value of unpaired one-sided $t$ test with $p < 0.05$ (*). $P$ values are 0.0056, 0.0314 and 0.0329 for 4 h, 6 h and 8 h, respectively. **d** Volcano plot representation of transcriptomics data (NLIM-PRO) of subpopulations that were separated based on pRPL28 marker intensity and cell size, as represented in **b**. $P$ values (two-sided) were adjusted for multiple hypothesis testing using the Benjamini–Hochberg procedure. Significant transcripts (indicated in green) are those with $p$ adj < 0.05 and absolute log2 fold-change (FC) ≥ 1.

after post-sorting resuspension in their original media composition (Supplementary Fig. 10). Sorted high fractions were able to regenerate the low subpopulation over time (Supplementary Fig. 10), further highlighting that the low subpopulation is a daughter-specific state arising from the high subpopulation. Once in the low state, the dynamics differed between NLIM-PRO and NLIM-GLN. In NLIM-PRO, most of the cells remained in this state and the mean subpopulation cell size and pRPL28 intensity slowly increased (Supplementary Fig. 10 and 11). Conversely, in the low NLIM-GLN subpopulation, the increase in cell size and pRPL28 intensity was more pronounced than for NLIM-PRO. Remarkably, the low NLIM-GLN subpopulation achieved similar levels in pRPL28 intensity and cell size to the high subpopulation 6 h post-sorting (Supplementary Figs. 10 and 11). To further validate our flow cytometry results, we also imaged and counted bud scars for each sorted subpopulation in each condition using calcofluor white staining (Fig. 3a). In both conditions, the low subpopulation primarily consisted of scarless daughters, whereas the high subpopulation was predominantly enriched in dividing mother cells (Fig. 3b). After 2 h regrowth in their respective media, we noticed again a significant difference between conditions. In NLIM-PRO, ~80% of the cells were daughters, whereas in NLIM-GLN, this number was only 30% (Fig. 3b).

This observation suggests that low subpopulation cells in the NLIM-GLN condition were resuming growth and regenerating daughter cells significantly more than in NLIM-PRO. Taken together, our results show that low subpopulations arise in the NLIM state and growth either stalls or resumes according to the nitrogen source present.

The cell size and pRPL28 fluorescence modality (i.e. unimodal or bimodal) observed above was conserved in nitrogen-replete conditions (10 mM), as well as intermediate concentrations, indicating that this was not a feature exclusive to nitrogen limitation but of the nitrogen source (Supplementary Fig. 12). We performed time-lapse microscopy of single-cells and could observe the emergence of three phenotypes in NLIM-PRO (Supplementary Fig. 13). Some cells (below 2% of the total population estimated from flow cytometry data; Supplementary Note 1) completely lost their fluorescence with cell size shrinkage, indicative of potential cell death, while most cells showed differences in pRPL28 intensity and cell size as observed during flow cytometry.

Finally, to assess if the observed heterogeneity was strain-specific, we analyzed the cell size modality in response to a downshift to NLIM-PRO or NLIM-GLN for 20 wild-types and laboratory *S. cerevisiae* and its close relative *S. paradoxus* isolated from diverse geographical

locations worldwide[40] (Supplementary Data 3). We found that, similarly to BY4741 and FY4 strains, multiple subpopulations (sometimes more than two) could be formed in NLIM-PRO and in NLIM-GLN among wild-type isolates (Supplementary Fig. 14), showing that cell size modality is not the result of domestication but suggests broader ecological implications inherited from wild-type strains.

## Quiescence heterogeneity is amino acid-specific

We next asked how a downshift to other nitrogen sources could affect quiescence heterogeneity. We performed a nitrogen downshift from rich media to 24 different nitrogen sources (20 amino acids and 4 non-proteinogenic sources) that can be transported across the plasma membrane[41]. For comparison, we also considered shifts to nitrogen-starved media (NSTARVE) and nitrogen-replete media (NREP, 10 mM) for a selection of nitrogen sources that were soluble under the conditions tested. Since cell size was an overall good discriminator of differentiation, we measured cell size heterogeneity 2 h, 4 h, 6 h and 8 h into the shift as a measure of quiescence heterogeneity. As seen previously for NLIM-PRO and NLIM-GLN, measuring cell size distribution 6 h into the shift was enough to assess the dynamics of the low subpopulation and capture differences between bimodal and unimodal conditions (Fig. 2a). Nitrogen downshifts gave a wide range of heterogeneity profiles over time for NLIM (Supplementary Fig. 15) and NREP (Supplementary Fig. 16). Among those, downshift to either ammonium, arginine, asparagine, methionine, or serine gave unimodal profiles over time, while downshift to other nitrogen sources or nitrogen-starved media gave bimodal profiles. Notably, a shift to glutamate gave a bimodal followed by a more unimodal profile after 8 h. Direct proximity to glutamine through the GS-GOGAT nitrogen assimilation pathway could explain this hybrid response. Glutamate (bimodal after 2 h), slowly converting to glutamine (unimodal), would indeed lead to a bimodal followed by a unimodal response. We further quantified the bimodality in cell size distribution for each of the 25 different nitrogen conditions (24 nitrogen sources and NSTARVE) using Hartigan's diptest bimodality scores (Methods) on three replicate flow cytometry experiments and found 14 NLIM conditions and 7 NREP conditions (out of 25 NLIM and 16 NREP tested, respectively) that gave significant ($p < 0.05$, Hartigan's diptest) bimodality scores 6 h into the shift for all replicates (Fig. 4a, Supplementary Figs. 17, 18 and 19a). We noted that many of the nitrogen sources leading to unimodal growth were part of the "preferred" nitrogen source group that triggers the Nitrogen Catabolite Repression (NCR) programme, usually associated with better growth performances[41,42]. This was the case for arginine, glutamine, ammonium, serine, and asparagine but not for alanine and aspartate, which trigger NCR but were bimodal. To further investigate the connection between modality and growth, we also recorded growth performance across all conditions (Fig. 4b, Supplementary Fig. 19b). Our results for NREP conditions showed a moderate-strong correlation (Spearman coefficient $\rho = 0.5–0.7$) with previous growth rate performance measurements on different nitrogen sources in *S. cerevisiae* (Supplementary Fig. 20). As expected based on the pRPL28 growth marker, nitrogen sources sustaining higher maximal growth rate had a unimodal distribution, while nitrogen sources with a bimodal distribution had slower growth, suggesting a strong anti-correlation (Fig. 4c; Spearman coefficient $\rho = -0.80$ and $-0.74$ for NLIM and NREP, respectively) between cell size bimodality and maximal achievable growth rate, further validating the computational growth prediction and pRPL28 growth marker identified from scRNAseq. Comparison between NLIM and NREP conditions show that NREP could sustain higher growth (Supplementary Fig. 21a) but did not show significant differences in modality across amino acids between NLIM and NREP (Supplementary Fig. 21b). Together, our data suggest that amino acid quality and not quantity is important for modality and that population unimodality correlates with growth maximisation. Remarkably, the presence of alternative nitrogen sources yielded more

extreme phenotypes than when nitrogen was absent. This includes a downshift to leucine which gave the most bimodal profile, or a downshift to either lysine or cysteine, which gave a significantly lower ($p < 0.05$ and $p < 0.0005$, respectively; unpaired $t$ test) maximal growth rate than when nitrogen was absent from the media (NSTARVE), further highlighting a strong connection between nitrogen sources and population dynamics.

## Heterogeneity follows a growth-viability tradeoff

Given the apparent growth defect associated with bimodal profiles, we asked what the evolutionary benefits of maintaining a low, quiescent subpopulation of smaller cell size could be. Previous research in *S. pombe* has shown that small-sized non-growing quiescent populations appear under nitrogen-starved conditions[43] with increased stress

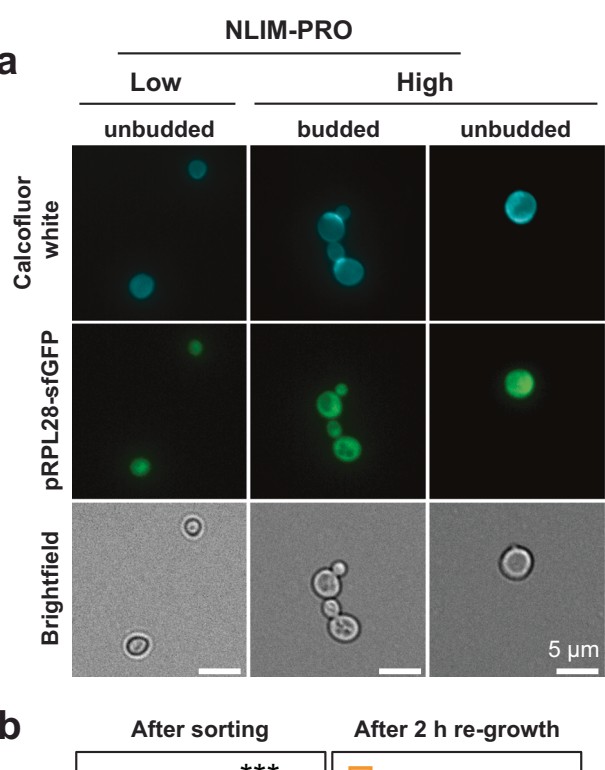

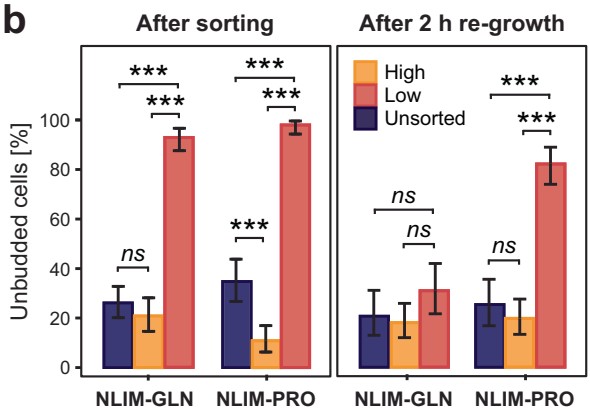

**Fig. 3 | The low subpopulation is originally enriched with daughter cells.** **a** Microscopy imaging of sorted fractions exposed to a 2 h downshift. Bud scars were visualised using calcofluor white dye on an inverted microscope with ×40 magnification. **b** Fraction of unbudded cells calculated by microscopy. Over 100 cells were imaged for each subpopulation and condition. Error bars represent 95%-confidence intervals centred on the individual fraction of unbudded cells. Significance scores denotes $p$ value with $p < 0.05$ (*), $p < 0.01$ (**), $p < 0.001$ (***) based on the Fisher's Exact test (one-tailed). Purple, light orange and light red bars represent unsorted (not gated), high and low fractions, respectively.

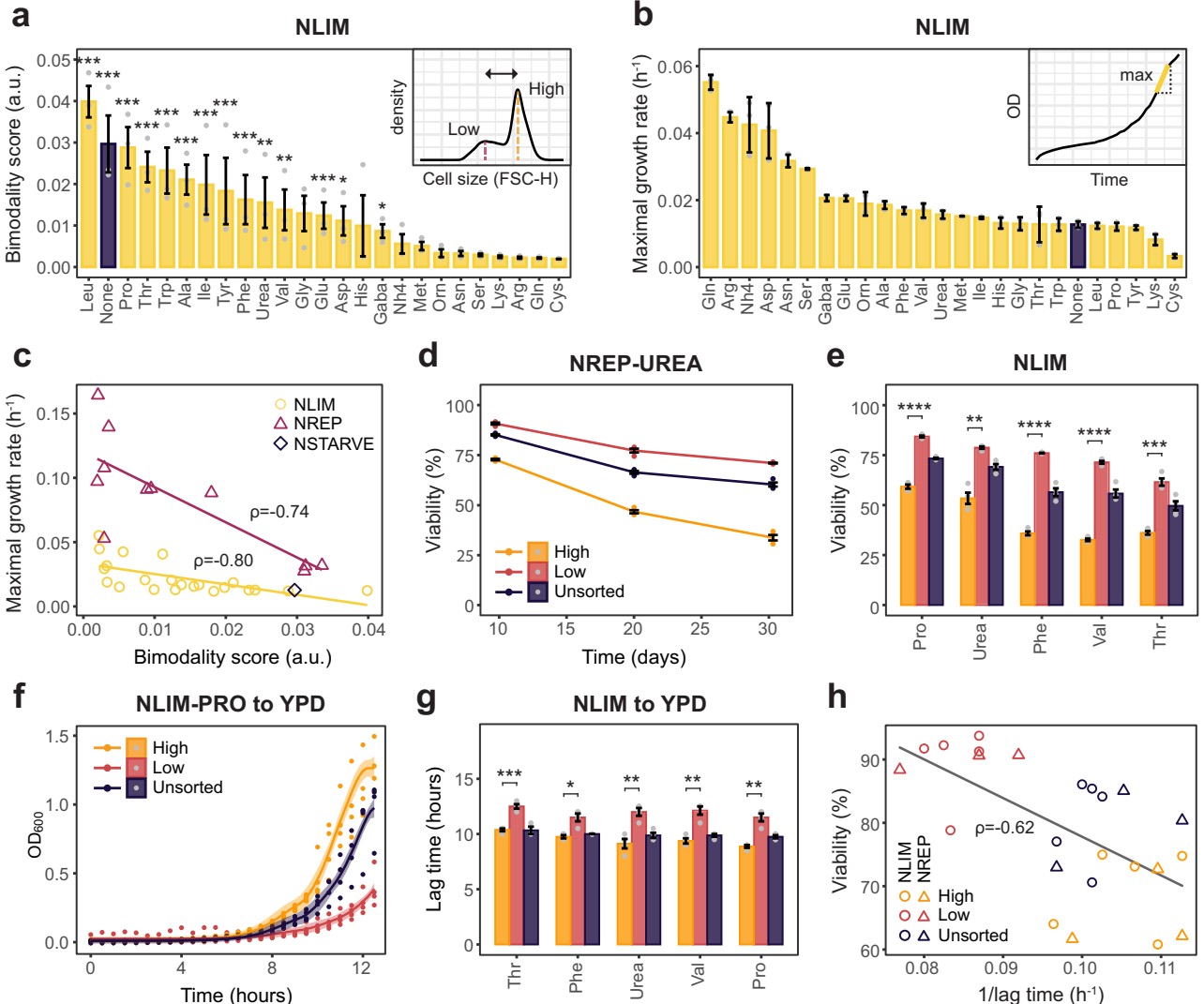

**Fig. 4 | Yeast subpopulations follow a growth-viability tradeoff that is nitrogen source dependent. a, b** Bimodality scores (Hartigan's diptest; Methods) and maximal growth rates across all NLIM conditions. Bar plot represents the mean ± SEM of **a** flow cytometry experiments for cells exposed to a 6 h downshift performed on different days ($n = 3$) and **b** growth curves of biological replicates ($n = 3$). Purple bars represent the NSTARVE condition where no nitrogen is present. Significance scores denotes conditions where all replicates had a $p$ value (Hartigan's diptest) of at most $p < 0.05$ (*), $p < 0.005$ (**) or $p < 0.0005$ (***). Bimodality scores are shown as arbitrary units (abbreviated a.u.). **c** Tradeoff between bimodality and growth, shown in **a**, **b**, respectively. Regression lines are indicated in yellow and magenta for NLIM and NREP, respectively. Spearman correlation scores are indicated for each regression line. NLIM-CYS and NLIM-LYS were omitted from the plot due to poor growth performances. **d, e** Viability rates for cells exposed to a 4 h downshift and sorted based on size and pRPL28 intensities using FACS. Yellow and pink samples indicate high and low GFP fractions. Dark purple samples indicate cells passed through FACS but not gated. Error bars represent the mean ± SEM ($n = 4$). Significance scores denotes p-value of unpaired two-sided $t$ test with $p < 0.05$ (*), $p < 0.005$ (**), $p < 0.0005$ (***) and $p < 0.00005$ (****). **f, g** Growth resumption in rich media (YPD) for sorted and unsorted fractions from **e. f.** **f** the shaded area represents the 95% confidence interval ($n = 4$) of the best fit using the ggplot2 function geom_smooth with "y ~ exp(x)" as fit function. **g** lag times were calculated as the time needed (discretized in 30 min interval) to achieve two doublings. Significance scores denote the $p$ value of paired two-sided $t$ test as in **e. h** Tradeoff between survival and growth resumption (measured as the inverse of the lag time). Black line represents the regression and the associated value the Spearman correlation. Circles and triangles represent NLIM and NREP conditions, respectively. **d–h** purple, light orange and light red colours represent unsorted (not gated), high and low fractions, respectively.

resistance[44]. With many subprocesses differentially regulated in quiescent cells linked to a general increase in survivability[4,9], we investigated links between bimodal growth and cellular viability. To this end, we exposed the cells to a 20 h downshift in each of the 24 nitrogen sources, washed them and stored them in PBS for 20 days. As a measure of cellular viability, we used propidium iodide staining, routinely used in chronological lifespan assays. Propidium iodide permeates non-viable cells, with the proportion of stained cells used to estimate single-cell viability and chronological lifespan. We discovered strong differences in viability across conditions with nitrogen sources sustaining unimodal and faster growth such as

glutamine, arginine, or ammonia generally displaying lower viability rates (Supplementary Fig. 22). NREP conditions lead to lower viability rates than NLIM showing that nitrogen source quantity could also influence viability (Supplementary Fig. 21c). To further assess if increased viability during bimodal growth could be tied to one of the subpopulations, we recorded subpopulation-specific viability rates. We exposed cells to a 4 h downshift (5 NLIM and 3 NREP different conditions) and subsequently sorted subpopulations based on cell size and pRPL28 expression. As control, we also kept an unsorted fraction that passed flow cytometry but was not gated. We observed the greatest difference in viability in NREP-UREA, where the viability rate

gradually decreased for all sorted and unsorted fractions over 30 days with the high subpopulation reaching a $33.7 \pm 1.4\%$ viability rate while, in comparison, the low subpopulation could maintain a remarkable $71.0 \pm 0.2\%$ viability rate (Fig. 4d). In fact, the low fraction could achieve significantly better viability rates and chronological lifespan than the high and unsorted fraction in all conditions tested (Fig. 4e).

We next sought to investigate how subpopulations would fare once optimal growth conditions would resume. To this end, we re-exposed each sorted fraction as well as the unsorted fraction to rich YPD media and monitored growth over 15 h. As a measure for growth resumption, we calculated a pseudo lag time, defined as the time it took to achieve two doublings (corresponding to a fourfold increase in $OD_{600}$). Remarkably, the high fraction could resume growth faster than the low and unsorted fractions. For NLIM-PRO, the high fraction achieved a lag time of $8.9 \pm 0.3$ h, while it took significantly longer ($p < 0.05$, paired $t$ test) for the low fraction to resume growth with a lag time of $11.5 \pm 0.7$ h (Fig. 4f, g). Again, the lag time extension observed for the low subpopulation was significantly longer than the high sub-population for all conditions tested (Fig. 4g), likely due to metabolic changes necessary to re-enter cell cycle progression. An extended lag time was also observed for the low subpopulation when grown in YPD on agar pads (Supplementary Fig. 23). Comparing viability and growth resumption rates, our data suggests a second tradeoff between long-term viability and growth maximization (Spearman correlation $\rho = -0.62$, Fig. 4g). In terms of population-wide resource allocation, investing in a subpopulation with higher viability and chronological lifespan is probably advantageous in fluctuating conditions and outweighs the associated population growth reduction. We indeed note that the low subpopulation investment seems to be mitigated during re-exposure to rich media, with the unsorted fraction showing a similar growth compared to the high subpopulation, especially under NREP conditions (Supplementary Fig. 24). Additionally, the increased lag phase in rich media for the low subpopulation could be a memory effect. Cells exposed to fluctuating environments often display history-dependent behaviours to prepare for recurring suboptimal conditions[45]. Therefore, maintaining a longer lag phase and not resuming growth straight after re-exposure to rich media could increase longevity as measured by viability extension and enhance survival chances, especially with reoccurring suboptimal conditions.

## Viability and growth are distinct transcriptional states

We further characterised metabolic differences between growth and viability based on transcription factor (TF) nuclear intensity, where TF activity is proportional to nuclear intensity in addition to promoter affinity of the downstream targets[46]. To this end, we created a proto-trophic transcription factor GFP fusion library (TF-GFP library) using Synthetic Genetic Array (SGA) (Methods). The library consists of 192 members containing a TF fused to GFP as well as a nuclear localisation tag (BFP) and a subpopulation marker (pRPL28-RFP). We performed a nitrogen downshift into proline or glutamine for each member of the TF-GFP library and analysed the resulting nuclear intensity over time for each condition and subpopulation using high-throughput microscopy (Fig. 5a). TF nuclear intensities were computed from YeaZ segmented cells by overlapping nuclear localisation tag (BFP) and TF-GFP localisation (GFP), while subpopulation assignment was performed on cell size and pPRL28-RFP intensity (Methods, Fig. 5a, b). For each GFP-tagged TF, we recorded cell size, mean cellular pRPL28-RFP and mean nuclear GFP for ~800,000 cells exposed to 30 min, 90 min, 150 min and 210 min of either NLIM-PRO or NLIM-GLN treatment (Source Data). The resulting cell size-pRPL28-RFP scatter plot obtained from microscopy was consistent with flow cytometry data (Fig. 5b, Supplementary Fig. 25). After 30 min, we found that already 157 and 116 transcription factors were significantly different ($p < 0.05$, two-tailed $t$ test) between conditions and subpopulations (NLIM-PRO), respectively (Fig. 5c, d). Because TF localisation can be noisy and highly variable over time[21], we

also ranked the nuclear intensities at each timepoint to highlight the most consistent TFs for each condition and subpopulation (Fig. 5e, f, Supplementary Data 4).

We first asked whether the TF library could capture established differences between proline and glutamine conditions. For example, Put3, activator of proline catabolic genes and under the control of the NCR programme[47], was one of the most localised TF in proline compared to glutamine at bulk and at the low subpopulation levels (210 min; $p < 0.05$, two-tailed $t$ test) (Supplementary Data 4, Supplementary Fig. 26a, b). Ure2, one of the most consistent targets over time, had a higher nuclear intensity for the proline condition at the population level (Fig. 5c, e). Ure2 is strongly associated with the NCR programme, sequestering the NCR activator Gln3 to the cytoplasm in presence of a preferred nitrogen source such as glutamine[48]. Dig2, a repressor of the Ste12 transcription factor, part of MAPK signalling, which was differentially activated between low and high subpopulation according to subpopulation RNA sequencing (Supplementary Fig. 9). To validate our high-throughput results, we further imaged Dig2-GFP strain during a shift to either NLIM-PRO or NLIM-GLN and could find it to be indeed more intense for proline (Supplementary Fig. 26f). Conversely, Mig1, involved in respiration and gluconeogenesis gene repression[49], had a higher nuclear intensity in glutamine, indicating that cells growing on glutamine could do fermentation despite nitrogen limitation (Fig. 5e, g). This was also conserved at the low subpopulation level where Mig1 was consistently more intense for glutamine compared to proline (Supplementary Fig. 26e). Consistent with this finding, we found that Snf1, a major Mig1 inhibitor, had higher nuclear intensity in the low proline subpopulation mimicking a low-glucose condition despite glucose present in abundance (Supplementary Fig. 26d)[50]. This corroborates our previous findings that the low glutamine subpopulation is actively growing while the low proline subpopulation is more quiescent.

We next used the TF library to investigate differences between high and low subpopulations in NLIM-PRO. Among the most consistent targets, Tec1 had significantly higher nuclear intensity in the low sub-population already after 30 min and onwards (Fig. 5d, f). We found that Tec1 was specific to the low proline subpopulation since it was more intense when compared to the low glutamine, indicating an essential role in maintaining quiescence (Supplementary Fig. 26c). Interestingly, Tec1 connects MAPK and TOR pathways to coordinate yeast development and has previously been reported as a positive regulator of chronological lifespan[51], consistent with the improved chronological lifespan observed for the low subpopulation. Tec1, together with Ste12 of the MAPK pathway, both regulate genes required for filamentous and pseudohyphal growth[52]. Rcs1, another TF that specifically localises to the low proline subpopulation, is involved in iron homoeostasis and its nuclear localisation increases with DNA stress[53] (Fig. 5f, Supplementary Fig. 26g). Other main targets included Rdr1 and Pdr8, both associated with ATP-binding cassette (ABC) family of transporters, important for resistance to drugs and other growth inhibitors and up-regulated in quiescent cells[9,54].

## Quiescent population is dormant and heterogeneous in ATP levels

Given important differences in transcription factor localisation for high and low subpopulations, we further studied how they could translate into physiological differences. Using metabolic sensors, we measured two key metabolic parameters, that are cellular energy status and metabolic activity. For energy status, we measured adenosine-triphosphate (ATP) levels using a FRET biosensor[55] (Fig. 6a). For metabolic activity, we measured fructose-1,6-bisphosphate (FBP), shown to correlate with glycolytic flux[5,56], using a riboswitch-based fluorescence sensor[57] (Fig. 6b). To account for growth differences between subpopulations as well as cell-to-cell variability in sensor expression that could occlude the measurement, FBP signal was

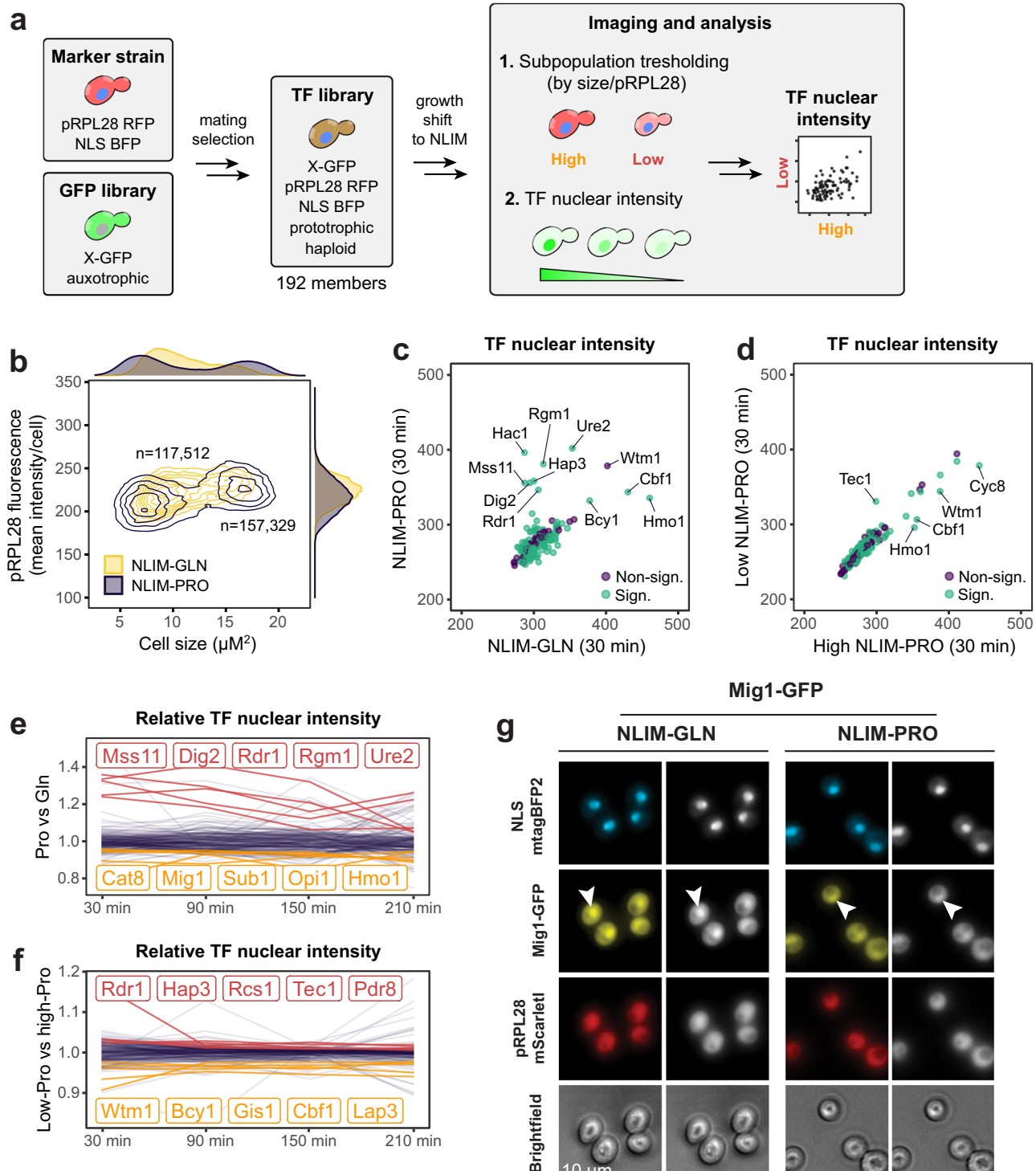

normalised to an RFP control placed under a constitutive promoter while the ATP sensor was ratiometric, providing the ratio between bound and unbound states (Fig. 6a, b).

Analysis of ATP levels after 8 h of growth in NLIM media with glutamine or proline showed population-wide differences and cellular ATP levels generally lower in glutamine than proline media (Fig. 6c). These data, together with the strongest growth profile observed in cells shifted to NLIM-GLN (Fig. 4b), could further suggest fermentative rather than respiratory growth in NLIM-GLN. Remarkably, while ATP levels were unimodal at the population levels, scatter projection of ATP levels versus cell size revealed cellular bimodality for proline, with

the emergence of two clusters corresponding to the low and high subpopulations (Fig. 6d). When we clustered high and low subpopulations based on cell size (Supplementary Fig. 27a), we found that from 4 h onwards, the low quiescent subpopulation showed a reduced but more heterogeneous ATP pool, while the high growing subpopulation maintained a homogeneous ATP content over the course of the experiment (Supplementary Fig. 27b). This difference in ATP heterogeneity between low and high subpopulations was maintained for the other bimodal nitrogen sources (Supplementary Fig. 28a). Similarly, conditions that sustained higher maximal growth rates (Fig. 4b) had generally lower median ATP output at the population

**Fig. 5 | Transcription factor tracking upon entry into quiescence. a** Overview of the pipeline. A marker strain containing the subpopulation marker as well as a nuclear localisation tag was mated with one of the 192 transcription factor members of the GFP library (TF-GFP) using the Synthetic Genetic Array method (SGA). The improved library was then grown in rich media and shifted to nitrogen-limited conditions as described in the Methods. The library was imaged in PBS 30, 90, 150 and 210 min into the shift. The YeaZ algorithm[26] was used to segment cells based on neural networks. When possible, segmented cells were assigned to high and low clusters based on pRPL28 mScarlet fluorescence and cell size using the Expectation Maximisation (EM) algorithm. TF localisations were computed for each cell and each strain by overlapping the nuclear localisation signal (mTagBFP2) with the TF-GFP signal (Methods). **b** Scatter plot representing mean RFP fluorescence per cell and cell size 150 min into the shift, at the onset of differentiation for NLIM-PRO

(dark purple) and NLIM-GLN (yellow). Cells were assigned to each cluster (low or high) based on their cell size and mean RFP intensity using the EM algorithm. **c, d** TF localisations for proline plotted against glutamine (population level) and high against low subpopulation (NLIM-PRO), 30 min into the shift. TF nuclear intensity was calculated as the mean GFP fluorescence over the nucleus, determined by the NLS-mtagBFP2. Significance score denotes *p* value of unpaired two-sided *t* test with *p* < 0.05, with an adjusted mean (condition/subpopulation difference). TF with significant localisation scores are shown in green. **e, f** Relative TF nuclear localisation tracked over time for proline versus glutamine (**e**) and low versus high proline subpopulations (**f**). Relative localisations were normalised by the mean relative localisation of the timepoint. **g** Multi-channel ×50 imaging of Mig1-GFP for one representative of two experiments. NLS-mtagBFP2 represents the nuclear localisation and pRPL28-mScarletI was used to classify high and low subpopulations.

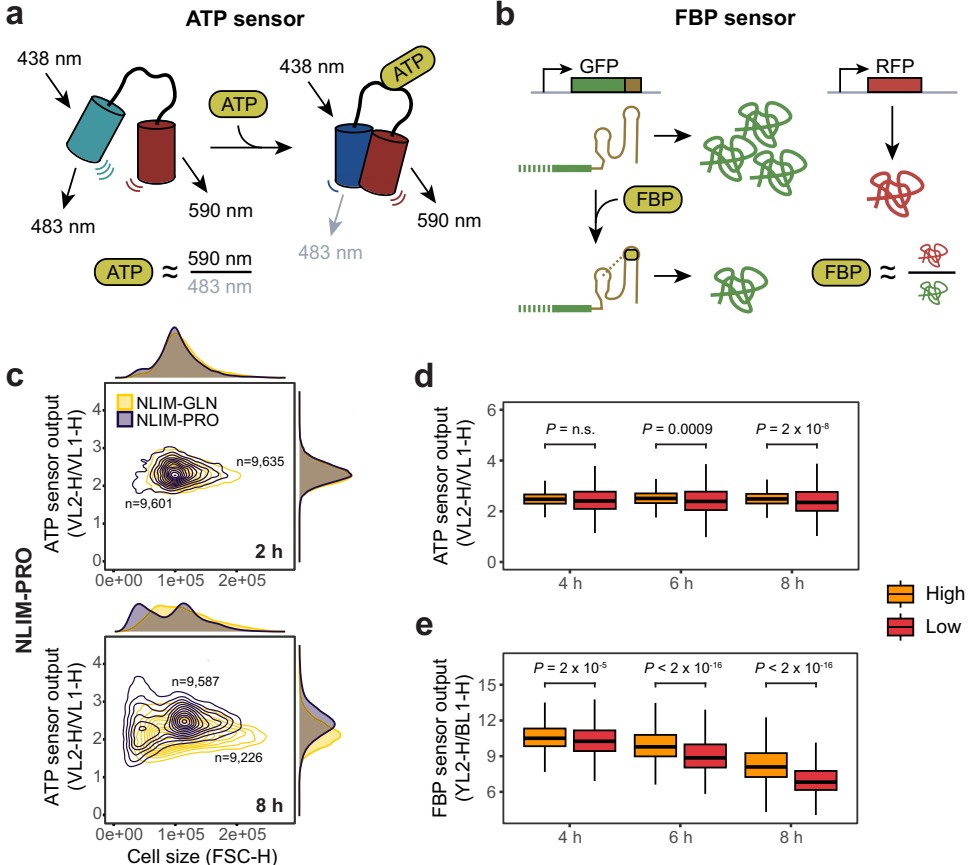

**Fig. 6 | Central metabolites tracking upon entry into quiescence. a, b** Overview of the ATP (yATP) FRET sensor[55] and FBP fluorescence sensor[57]. Both sensors are ratiometric, normalising out potential differences in sensor expression levels. For the ATP sensor, relative ATP levels are proportional to the inverse of the ratio of acceptor (590 nm) over donor (483 nm) emission from the same excitation (438 nm). For the FBP sensor, FBP levels can be estimated from the inverse of the ratio of GFP (riboswitch) over RFP (control) ratio. GFP and RFP are integrated in the same locus (URA3) and under the control of pTEF1 and pTEF2, respectively. **c** Scatter plot representing the emergence of the low subpopulation for NLIM-PRO

(dark purple) and NLIM-GLN (yellow) based on the output of the ATP sensor (yATP) against cell size. **d, e** Quantification over time based on Expectation Maximisation clustering (Methods) of scatter plots as shown in **c**. Orange and red colours represent high and low fractions, respectively. Only timepoints where clustering could be performed are shown. Centre marks of the box plot represent the median, hinges mark the lower and upper quartiles and whiskers show all values that, at maximum, fall within 1.5 times the interquartile range. Indicated p-values were calculated from unpaired two-sided *t* test and represent single-cell heterogeneity performed on one representative of two experiments.

level (Supplementary Fig. 28b, c). A high variability in the quiescent subpopulation could be consistent with bet-hedging where increased ATP variability could denote a strategy to further increase phenotypic diversity in preparation to environmental fluctuations. Similar observations were made for cells shifted from glucose to maltose, where cells resuming growth had a higher and more homogeneous ATP pool than cells that remained quiescent[29]. Conversely, the analysis of FBP levels did not show the same difference in variability between low and

high subpopulations but showed a population-wide decrease over time instead (Fig. 6e). This decrease was more marked for the low subpopulation suggesting a more dormant metabolism, consistent with previous studies on quiescence[6,58].

## Discussion

Our results show that *S. cerevisiae* exposed to a nitrogen downshift display phenotypic bimodality with simultaneous presence of

subpopulations following a tradeoff between chronological lifespan and growth capabilities. Concomitantly, these subpopulations operate at very distinct metabolic states where the low subpopulation is optimized for viability, and the high subpopulation is optimized for growth. With these fundamentally opposite metabolic states, individual cells need to commit to either state (specialisation) and phenotypic bimodality is therefore optimal given finite resource allocation. A similar phenotypic specialisation into a growing and a stress-resistant fraction has also been found when *S. cerevisiae* was exposed to growth inhibitors[35]. In proline, the low subpopulation is a terminal state enriched with daughter cells, while in glutamine, it can resume growth. Asymmetric generation of quiescent-specific daughter cells has also been observed during glucose exhaustion, at the onset of stationary phase[10]. Using subpopulation-based RNA sequencing and high-throughput TF imaging, we found that increased viability correlates with a more active MAPK signalling and DNA stress response associated with general chromosomal regulation, as well as organelle and morphological reorganisation. The seemingly more active MAPK signalling between subpopulations denotes an important crosstalk between nitrogen and carbon metabolism. More importantly, the low subpopulation had similarities with a low-glucose state despite glucose present in excess. Nuclear localisation of transcription factors involved in metal homoeostasis for the low proline subpopulation could be related to the non-growing phenotype, where upregulation of ion homoeostasis when growth stalls have been previously reported in a wide range of model organisms[9,59,60].

We argue that in the context of evolution and given conservation in both wild and domesticated *S. cerevisiae* and *S. paradoxus* strains, cell size bimodality in response to a nitrogen downshift is a hard-wired regulatory mechanism and probably crucial to environmental interactions. We further hypothesise that amino acids could be used as an environmental cue for decision-making. In this aspect, glutamine could denote ideal growth conditions while leucine, which gave the strongest cell size bimodality (Fig. 4a), could indicate conditions that necessitate a survival increase or dispersal. Supporting this theory, cells growing on leucine produce a distinctive banana-like aroma including isoamyl alcohol, isovaleric acid or isoamyl acetate[61]. While aroma production dependence on nitrogen sources is well documented in the yeast brewing industry[62], its ecological purpose still remains obscure but past evidence suggests that aroma production could be used for attracting insects and facilitating dispersal[2,63,64]. Strong differentiation to quiescence for leucine coupled with increased aroma production could, therefore, facilitate dispersal through insect vectors directly linking the observed bimodality to a broader ecological context. Given that growth on leucine was more bimodal than growth without any nitrogen sources (NSTARVE), we further hypothesise that leucine could be used as an environmental cue for dispersion. Increasing viability and extending lifespan might be a strategic choice to allow better dispersion without needing to undergo sporulation and could offer several advantages. First, it could be seen as a lighter, more reversible commitment to dormancy. This could explain why diploid cells in the presence of fermentable carbon sources such as glucose undergo quiescence rather than sporulation when other essential nutrients such as nitrogen are limiting growth[65]. Second, the increased lag phase for the quiescent subpopulation could correspond to a delayed phenotypic outing and give a memory effect in fluctuating conditions to better withstand previously experienced suboptimal conditions[45,66]. We note that this is still unclear whether the increased lag phase could also be a by-product of increased viability where cells need time to exit quiescence. Yet, another advantage could be that quiescent cells remain metabolically active and could still be able to exchange metabolites with their environment. Metabolite exchange, amino acids in particular, is a key feature of yeast exo-metabolome and is important for phenotypic heterogeneity and metabolic specialisation[67]. Remarkably, while measuring the supernatant of yeast exposed to 8 h nitrogen downshifts, we could detect amino acids other than the nitrogen source used, even under low nitrogen contents (NLIM, 0.8 mM) (Supplementary Data 5), supporting a broader population-wide function based on amino acid exchange. Similarly, a recent study found that lysine was involved in cross-feeding interactions in ageing colonies on an agar plate resulting in phenotypic heterogeneity between young and old cells consuming and producing lysine[68]. Interestingly, we found that lysine was the amino acid that performed the worst in both growth and chronological lifespan. We could not detect any lysine throughout the different downshifts despite the observation that *lys12* (lysine biosynthesis) knockout is able to establish syntrophic communities with lysine producers[69], showing that lysine or its intermediate can be exchanged.

In light of a recent study showing that artificial consortia of *S. cerevisiae* auxotrophs have an extended chronological lifespan[70], we show that such lifespan improvements could be explained by the heterogeneity we observed. Auxotrophs would typically be forced to rely heavily on amino acids produced by other members of the community and could, therefore, exhibit heterogeneity within an auxotrophy class, leading to increased lifespan. Finally, in terms of fitness burden, our results show that an investment in a small quiescent subpopulation with longer lifespan is beneficial since it results in a minor decrease in growth resumption capabilities with the lag phase for high and unsorted subpopulations being very similar if not the same in certain cases. This can explain other bet-hedging strategies such as antimicrobial resistance[71]. Our results could be relevant in future work to study amino acid sensing mechanisms or investigate potential division of labour across subpopulations and the role they play within a natural environmental context.

## Methods

### Single-cell RNA sequencing data analysis

Single-cell RNA sequencing data were obtained from ref. 23. Raw counts data was downloaded from NCBI with accession number GSE125162. All data analysis was performed on R. Pre-processing was performed according to the author's guidelines, except for the normalisation that was performed using the logNormCounts from the LTLA/Scuttle package (https://github.com/LTLA/scuttle). Dimensionality reduction using Uniform Manifold Approximation and Projection (UMAP) was performed using the scater package[72]. To calculate single-cell growth scores, the log normalised scRNAseq reads were inputted into the calculateRates function from the growth regression model obtained from ref. 39. To identify subpopulation-specific markers, DESeq2[73] was used to find genes that were differentially expressed between subpopulations. Detailed methodology is available as part of Supplementary Method 1. All data analysis performed in this study are available on GitHub (https://github.com/KiyanShabestary/2023-NLIM-heterogeneity).

### Strain creation

Two prototrophic strain versions based on the laboratory strain BY4741 (*MATa his3Δ1 leu2Δ0 met15Δ0 ura3Δ0*)[74] were made as subpopulation marker strains. One version had full prototrophy restored using the minichromosome pHLUM series[75]. A genome-integrated version was also created with HLUM fragments, PCR amplified from the respective minichromosomes with ~35 bp overhangs for homologous recombination targeting the HO locus. The fluorescent protein sfGFP was assembled under the control of the pRPL28 promoter, identified as a subpopulation marker through scRNAseq data analysis. Transcription termination was under the control of the tTDH1 terminator. The pRPL28 promoter part was amplified from genomic DNA taking 700 bp directly upstream of the coding sequence. Parts for sfGFP and terminators were obtained from the yeast MoClo Toolkit (YTK)[76]. Assembly was performed into a URA3 targeting vector using Golden Gate assembly as described in the YTK toolkit assembly. For

the ATP sensor strain, a codon-optimised variant of yAT1.03 sensor[55] was ordered as GeneArt from ThermoFisher and cloned in a pYTK level 0 Type 3 plasmid. Using Golden Gate assembly, the fragment was placed under pTEF1 control and with the tENO1 terminator in a vector targeting *ura3*. The plasmid was co-transformed with pHLM into BY4741. For the FBP sensor, the 2_riboswitch sensing unit was amplified from the pFBP-2_6.sensor[57] plasmid obtained from addgene (catalogue number 162800). The fragment was assembled into pYTK053 of the YTK collection, downstream of the promoter pTEF1 and mNeonGreen and upstream of the tSSA1 terminator. Both the FBP sensor and its control (mScarletI (RFP) under pTEF2 control and with tENO2 terminator) were genome-integrated at the *ura3* locus, complemented with the pHLM minichromosome. For the TF library, a donor strain based on BY4742 (MATα *hisΔ1 leu2Δ0 lys2Δ0 ura3Δ0*) was used to create the prototrophic GFP library using SGA (see SGA section in Methods). As a subpopulation marker, mScarlet (RFP) was placed under pRPL28 control at the *ura3* locus with tTDH1 terminator. For nuclear localisation visualisation, a nuclear localisation tag derived from the SV40 T-antigen[77] was added at the 5′ end of mTagBFP2 (BFP) placed under the control of pTEF2 and with the tENO2 terminator genome-integrated at the *leu2* locus. Additionally, for haploid selection, a kanMX cassette was placed under the MATa-specific pSTE2 promoter (genome amplified) with tPGK1 terminator at the *can1* locus. All assemblies were performed using Golden Gate assembly within the YTK framework[76]. Strains used in this study are indexed and described in Supplementary Data 3.

## Yeast transformation

Transformation into yeast was performed using the Lithium acetate protocol[78]. Overnight YPD yeast cultures were diluted in YPD (1:50, 5 ml per three transformations) in the morning and cultivated until they reached exponential growth (4–5 h). Cells were then washed once and resuspended in 0.1 M Lithium acetate (LiOAc, Sigma) to a final volume of 100 µl per transformation. The subpopulation marker plasmid was linearised using NotI (New England Biolabs). The linearised plasmid (500 ng) and the minichromosome (1 µl, >100 ng/µl), when appropriate, were mixed with boiled (5 min, 100 °C) salmon sperm DNA (10 µl, Invitrogen). Competent yeast cells were resuspended in the DNA/salmon sperm DNA mixture and then mixed with 260 µl 50% (w/v) PEG-3350 (Sigma) and 36 µl 1 M LiOAc. The transformation mixture was incubated at 42 °C for 25 min, resuspended in sterile water and plated on the appropriate selection medium. All strains were confirmed by cPCR followed by Sanger sequencing.

## Growth conditions

Yeast cells were cultivated in sterile 14 ml cell culture tubes (Greiner Bio-One) grown at 30 °C in a Infors HT Multitron with 700 rpm shaking. Yeast extract-peptone dextrose (YPD) composed of 1% (w/v) Bacto Yeast Extract (Merck), 2% Bacto Peptone (Merck), and 2% Glucose (VWR) was used as rich pre-shift media. For the post-shift media, 1.7 g/l Yeast Nitrogen Base (YNB) without amino acid and ammonium sulfate (Sigma) and with 2% Glucose (YNB) was used with 0.8 mM (NLIM), 10 mM (NREP) or without (NSTARVE) nitrogen source. All amino acids were supplied from Sigma or Formedium. Both YPD and post-shift media were buffered with 50 mM phosphate buffer and adjusted to pH 6.0.

## Medium shift

Pre-shift growth included an overnight step in YPD from a single colony stored on YNB agar plate followed by a 1:50 dilution in 5 ml YPD grown for 4 h until the end of exponential growth was reached ($OD_{600}$ = 0.8–1.0). Cells were then centrifuged for 6 min at 4 kG and washed in phosphate-buffered saline (PBS) solution twice. After the washing step, cells were resuspended in post-shift media (NLIM, NREP, or NSTARVE) to $OD_{600}$ = 0.4 in 14 ml cell culture tubes or 250 ml flasks

(for subpopulation RNA sequencing) and grown at 30 °C with 700 rpm shaking (Infors).

## Time-lapse microscopy

Time-lapse microscopy was performed using agarose pads. In short, cells were trapped in between a microscope slide and an agar pad as previously described[79]. Agarose pads were composed of the respective growth media and 1.5% (w/v) low melting agar (Sigma). To create the agar pad, 1 ml of agar/media mixture was pipetted on top of a microscope glass cover (22 mm × 22 mm, VWR). Another cover was placed on top to create a layer of even thickness. Approximately 50 mm² of the solidified agarose (one-ninth) was cut out to make an agarose pad and 2 µl of cells ($OD_{600}$ = 0.4 in PBS) was applied in the middle of each pad. The agarose pad was then placed upside down in an enclosed 35 mm cell imaging dish (Ibidi). Water was added to the enclosure to limit evaporation during the time-lapse. Imaging was performed on a Nikon Ti-2 Twin-Cam-TIRF with an environmental chamber to maintain temperature at 30 °C.

## Subpopulation sorting

Subpopulations were sorted using fluorescence-activated cell sorting (FACS), performed on a BD FACSAria III Cell Sorter, based on GFP fluorescence (FITC-A, blue laser 488-530/30 nm) and morphology (SSC-A). Doublets and budding yeast exclusion were filtered out through FSC-W/FSC-H and SSC-W/SSC-H gatings. Prior to sorting, samples were filtered, and 20,000 events were used to adjust gating. Purity cheques were performed at the start of every sorting run to ensure accurate gating and no cross-contamination. Gating details are available as part of Supplementary Note 1. For subpopulation RNA sequencing, pRPL28 marker strain culture grown overnight was diluted (1:50) in YPD (10 ml) and grown to exponential phase (4 h). Cells were washed twice with PBS as described above and exposed for 1 hour in post-shift media (post-shift resuspension $OD_{600}$ = 0.4). Prior to sorting, cells were centrifuged, resuspended in PBS and kept at 4 °C for the duration of sorting. After sorting in 15 ml tubes, cells from each sorted fraction were grouped and centrifuged (6 min at 4 kG) to remove the supernatant. Cell pellets (at least 4 million cells for each sample) were frozen in liquid nitrogen and stored at −80 °C until RNA extraction. Growth resumption and chronological lifespan experiments were performed in 96-well plates (Greiner). Cells were grown in YPD and exposed to a 4 h post-shift as described above. Cells were washed once in PBS and sorted in 15 ml tubes. Approximately 50 k cells were used per well for both subpopulation lifespan and growth resumption measurements.

## Subpopulation RNA sequencing

Subpopulation RNA extraction, sequencing and data analysis was performed through Novogene sequencing services. Sorted cell pellets stored at −80 °C were thawed on ice. RNA was extracted with a RNA-prep Pure Plant Plus kit (Tiangen). Messenger RNA was purified using poly-T oligos attached magnetic beads. After fragmentation, the first strand cDNA was synthesised using random hexamer primers, followed by a second strand cDNA synthesis. The library was quantified using real-time PCR and Qubit. Size distributions were calculated using Bioanalyzer analysis. Quantified libraries were pooled and sequenced on an Illumina platform (Novaseq 6000) with a paired-end 150 bp (PE150) method. Raw sequencing reads and count matrix are available in NCBI GEO under the accession number GSE235239.

RNA sequencing reads were filtered according to the following criteria: Reads with no adaptor contamination, no more than 10% of uncertain base (N) within the read, not >50% of the reads made of low-quality base reads (Base Quality Qscore less than 5). Low-quality reads represented less than 1% of total reads. HISAT2[80] was used to map the filtered reads to the genome. Reference genome (fasta file) and gene annotations (gtf file) used for alignments were obtained from the

Ensembl database available at http://ftp.ensembl.org/pub/release-75 (December 2022).

Principal Component Analysis was performed on gene expression values (FPKM) to evaluate intergroup and intragroup variance and remove outliers. Differential gene expression was computed using DESeq2[73] and $p$ values adjusted using the Benjamini–Hochberg procedure. KEGG enrichment analysis was performed using clusterProfiler[81] with adjusted $p$ values obtained from DESeq2.

## Growth measurements

Growth parameters were calculated for cultures grown in YPD and shifted to post-shift media ($OD_{600} = 0.1$) as described above. Growth curves were obtained for 96-well plates (Greiner) recorded in a Tecan Spark microplate reader set at 30 °C with 200 rpm orbital shaking. Breathe-Easy sealing membrane (Sigma) was applied to reduce evaporation while maintaining gas transfer throughout the experiment. Bulk population maximal growth rates were calculated as follows: Data from the plate reader was blank normalised. The R package growthcurver (https://github.com/sprouffske/growthcurver) was used to obtain a smooth fit of the growth curve. Local growth parameter mu_log for each time interval was computed as the difference of the natural logarithmic of the ODs divided by the time interval. Finally, the maximal value mu_log within a time window excluding lag and stationary phases (between 5 h to 20 h in post-shift media) was taken as maximal growth rate. For growth resumption experiment in YPD following FACS-based sorting, cultures were resuspended to $OD_{600} = 0.05$. Lag time was estimated as the time required (discrete interval) to reach two doublings ($OD_{600} = 0.2$).

## Bimodality quantification

Cell-to-cell heterogeneity of cultures grown in YPD and shifted to post-shift media ($OD_{600} = 0.4$) was measured over time using an Attune Nxt flow cytometer (Invitrogen). Forward scatter (FSC-H) and GFP fluorescence (BL1-H, blue laser 480/10 nm) were used to measure cell size and GFP fluorescence heterogeneity, respectively. Doublets were excluded from the data based on FSC-H/FSC-A linear correlation (both log scale) using the FlowJo software. At least 10'000 events were recorded per sample/replicate. Cell size bimodality scores were computed using Hartigans' diptest for unimodality using the R package diptest (https://github.com/mmaechler/diptest). The stat and pval output values were used as bimodality score and associated p-value. The R package flexmix[82] was used to perform Expectation Maximisation on cell size distributions to fit a Gaussian model for each subpopulation and estimate the respective mean and standard deviation for each subpopulation.

## Chronological lifespan assay

Chronological lifespan was estimated as the percentage of cells remaining viable in PBS over time. For bulk population lifespan estimation, cultures (150 µl) exposed to a 18 h nitrogen downshift in 96-well plates were washed twice in PBS, resuspended in 200 µl PBS and 50 µl of cells/PBS mixture was added to 150 µl PBS in a 96-well plate. For subpopulation lifespan assessment following FACS-based sorting, cells exposed to a 4 h downshift were washed prior to sorting and 50 k cells in PBS were stored in 96-well plates (200 µl). One whole plate was used for each timepoint measurement to limit subsequent evaporation. Lifespan was measured up to 30 days after the end of the post-shift as indicated in text. Plates were sealed with Breathe-Easy sealing membrane (Sigma) and wrapped with aluminium foil and stored at 30 °C until cell viability measurement. Cell viability was measured on the basis of permeability to propidium iodide (PI, Merck) in apoptotic cells using flow cytometry (Attune Nxt, YL1-H, excitation 561 nm, emission filter 585/16 nm). Prior to the fluorescence assay, 1 µl of PI (1 mg/ml) was added to 200 µl cell/PBS mixture and gently mixed using a multi-channel pipette. For each sample/replicate, viability was thresholded based on viability measured in YPD.

## Unbudded cells counting

Sorted fractions of cells exposed to a 2 h downshift and sorted via FACS were collected in PBS and either directly stained or transferred to fresh NLIM-PRO or NLIM-GLN media and incubated for 2 h at 30 °C and 700 rpm shaking before staining. Bud scars were stained using calcofluor white (Sigma-Aldrich) at a final concentration of 0.01 g/L. Cells were incubated for 15 min in the dark at room temperature and washed in PBS. Subsequently, 5 µl of cells were transferred to microscopy glass slides and imaged using a Nikon Ti microscope fitted with a Hamamatsu Flash 4 camera. Stained bud scars were visualised using a P4000 Cooled LED light source at 365 nm and filters for blue fluorescence. pRPL28-sfGFP fluorescence was captured using a 460 nm LED and green fluorescent filters. Counting was performed manually, to prevent bias due to differences in staining efficiency.

## Metabolic sensors analysis

Metabolic sensors were used for in vivo measurement of ATP and FBP. Both sensors were genome-integrated at the URA3 locus. We used a FRET-based biosensor[55] for ATP and a riboswitch-based fluorescence sensor for FBP measurement[57]. For the ATP sensor, the ATP FRET signal was recorded using a 405 nm excitation laser, a 450/40 nm donor emission filter and a 525/50 nm acceptor emission filter (VL1 and VL2 channels). ATP levels were calculated by taking the ratio between the VL2 and VL1 channels (VL2-H/VL1-H). For the FBP sensor, the FBP signal was recorded using a 488 nm excitation laser and a 530/30 nm emission filter (BL1 channel) while the control RFP signal was measured using a 516 nm excitation filter and 620/15 emission filter (YL2 channel). With the FBP sensor fluorescence displaying a signal inversely proportional to FBP concentration. FBP levels were calculated by taking the ratio between the YL2 and BL1 channels (YL2-H/BL1-H). Signals were recorded on a Attune Nxt flow cytometer (Invitrogen). Expectation maximisation using the R package Rmixmod[83] with "Gaussian_pk_L_I" model selection was performed on cell size to compute subpopulation-specific ATP and FBP levels.

## Library creation using SGA

A prototrophic version of the GFP collection[84] was created using the Synthetic Genetic Array method[85]. A transcription factor library (TF-GFP library) containing 192 members was created as follows. Selected strains from the GFP collection (MATa his3Δ1 met15Δ0 leu2Δ0 ura3Δ0 XXX-GFP-HisMX) were mated with a donor strain based on the laboratory strain BY4742. The donor strain contained a nuclear localisation marker subpopulation marker as well as kanamycin resistance placed under the MATa-specific pSTE2 promoter for haploid selection (MATα can1::pSte2-KanMX-tPGK1 his3Δ1 lys2Δ0 leu2::pTEF2-NLS-mtagBFP-tENO2-LEU2 ura3::pRPL28-mScarletI-tTDH1-URA3) (see Strain creation method section above). Mating was performed on solid agar plates in a 384 array using a Rotor pinning robot (Singer Instruments). Cells were then transferred to agar plates with minimal media (YNB) lacking amino acids to select prototrophic diploids. These diploids were then incubated in pre-sporulation media (YP with 1% potassium acetate) in liquid 96-well plates and grown for 24 h. Cells were then washed in PBS and resuspended in 1% potassium acetate sporulation media. After 5 days, haploid MATa spores were selected by transferring 50 µl of spores into 450 µl of synthetic media containing 50 µg/ml canavanine and 300 µg/ml G418. After 24 h, 50 µl were washed in PBS and resuspended in 450 µl of synthetic media lacking uracil, leucine, histidine, lysine and methionine containing 10 µg/ml canavanine and 300 µg/ml G418 to select prototrophic haploids overnight. This step was repeated in standard 96-well plates in total volumes of 150 µl and single colonies were selected by transferring cells onto rectangular agar plates using the Rotor pinning robot

(Singer Instruments) and a $7 \times 7$ pinning protocol to select clonal populations from single colonies. Correct ploidy was confirmed using MATa- and MATα-specific primers as described in ref. 86.

## High-throughput microscopy
The transcription factor library consisting of 192 members was distributed along a 384 well plate μClear flat-bottom (Greiner) and imaged in a Nikon Ti-2 Twin-Cam-TIRF. Two images were taken at two different locations per library member using a bright field, blue, green and red fluorescent filter. To obtain maximal resolution without oil appliance, an ×40 optic with an additional ×1.5 lens was used. Looping through each unique well, an nd2 file was generated with a dimension of (fov(384) × channels (4) × Z stack (1) x X dimension x Y dimension)).

All data analysis was performed through a custom Python pipeline available on GitHub (https://github.com/Benedict-Carling/YeaZ-Output-Analysis). The generated nd2 file was used as an input for segmentation using the segmentation tool Yeast-Analyzer (YeaZ)[26]. The bright field segmentation parameters with a minimum seed distance of 1 and a threshold value of 0.5 were used. A scatter graph was generated for each cell identified by YeaZ with the $x$ axis representing the cell size and the y-axis representing the mean Red fluorescence of the cell. To mitigate the influence of outliers, such as missegmentations in the scatter plot, we employed the KernelDensity utility from the Scikit-learn Python package[87] with the following arguments: algorithm was set to ball_tree, bandwidth set to 1, metric set to Euclidean and kernel set to linear. Expectation Maximisation (EM) was performed using the Gaussian Mixture utility from Scikit-learn on the filtered cells to assign cells to each subpopulation cluster. We performed EM with a confidence threshold of 0.85 to remove any manual intervention in the identification of the subpopulations.

To generate nuclear segmentation, the bounding box of each cell as identified by YeaZ was looped through using the blue channel (nuclear localisation marker). The image was smoothed using a Gaussian blur of sigma = 1. A mask of the top 15% brightest pixels was generated, and erosion and dilation were performed to remove isolated islands, returning a mask representing the nucleus of each cell. Single-cell transcription factors nuclear intensity were calculated by averaging the GFP signal over the nuclear mask. Subpopulation-specific scores were obtained by averaging the scores of each single-cell for a given subpopulation. For condition-specific scores, localisation scores were then averaged across subpopulations. To obtain the most consistent transcription factors over time, relative TF scores were computed at each point. For the relative TF score calculation, each TF score was normalised by the mean of all TF that were not part of the top 5 or bottom 5 transcription factor at a given timepoint.

Targets identified in the previous step were validated using inverted microscopy using a Nikon Ti (×60 magnification). Exposure time was kept constant for each channel (200 ms for RFP, 500 ms for BFP and 1 s for GFP) with 11 slices per z stack to capture the nucleus. Cells were exposed to a 30 min shift prior to imaging.

## Statistical analysis and reproducibility
Heterogeneity in cell size and GFP measurements using flow cytometry were performed in triplicates and on separate days. Chronological lifespan measurements were performed at least on three biological replicates. Paired statistical analysis was performed using the $t$ test function (unpaired, two-sided) in R. Statistical analysis of bimodal distributions was performed using Hartigan's diptest. Significance scores for genes differentially expressed during subpopulation RNA-seq and single-cell RNAseq were adjusted using DESeq2 (multiple hypothesis adjusted $p$ value, Benjamini–Hochberg procedure). For significance testing of discrete data obtained from manual budscar quantification, we used Fisher's exact testing, and error bars represent 95% confidence intervals.

## Reporting summary
Further information on research design is available in the Nature Portfolio Reporting Summary linked to this article.

## Data availability
Data for Fig. 1c was from ref. 23 obtained from NCBI with accession number GSE125162. Raw sequencing data of subpopulation RNA sequencing generated in this study was deposited in NCBI GO with accession number GSE235239. All data generated or analysed during this study are included in this article and in the Supplementary Information. Source data are provided with this paper.

## Code availability
All scripts used for data analysis and plotting are available on GitHub (https://github.com/KiyanShabestary/2023-NLIM-heterogeneity for general analysis and plotting and https://github.com/Benedict-Carling/YeaZ-Output-Analysis for high-throughput microscopy).

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

## Acknowledgements

We thank Davina Patel (RLA lab) for helping with flow cytometry, Maria Portella (RLA Lab) for helping with the sensors, Larissa Zarate Garcia and the Imperial College Flow Cytometry Facility (Sir Alexander Fleming Building) for helping with cell sorting, David Bell (SynbiCITE) for helping with metabolomics, Mary Dunlop (Boston University) for fruitful discussions regarding cell tracking, Peter Thorpe (Queen Mary University) for providing members of the yeast GFP collection, Gianni Liti (Université de Nice) and Lars Steinmetz (EMBL Heidelberg) for providing laboratory as well as WT *S. cerevisiae* and *S. paradoxus* isolates. RLA received funding from BBSRC (BB/R01602X/1, BB/T013176/1, BB/T011408/1—19-ERACoBio-Tech- 33 SyCoLim), EPSRC (AI-4-EB), Yeast4Bio Cost Action 18229, European Research Council (ERC) (DEUSBIO—949080) and the Bio-based Industries Joint (PERFECOAT—101022370) under the European Union's Horizon 2020 research and innovation programme. The Facility for Imaging by Light Microscopy (FILM) at Imperial College London is partially supported by funding from the Wellcome Trust (grant 104931/Z/14/Z). K.S. acknowledges a postdoctoral fellowship from the European Molecular Biology Organisation (EMBO) (ALTF 769-2021) and a UKRI-Marie Skłodowska-Curie Actions (MSCA) Postdoctoral Fellowship (UNICOH).

## Author contributions

K.S. and R.L.A. conceptualised the study. K.S., C.K., B.C., and J.M. performed the experiments. C.K. and K.S. designed and created the TF-GFP library. B.C. and K.S. designed and created the high-throughput microscopy pipeline. J.S. cloned the metabolic sensors. K.S., C.K., and B.C. performed data analysis. K.S. wrote the draft. K.S., R.L.A., and M.S. supervised. All authors edited and reviewed the final manuscript.

## Competing interests

The authors declare no competing interests.
