## [Peer Review File · Nature Communications]

Phenotypic heterogeneity follows a growth-viability tradeoff in response to amino acid identityREVIEWER COMMENTS

Reviewer #1 (Remarks to the Author):

Shabestary et al. present a manuscript titled "Phenotypic heterogeneity in lifespan follows an evolutionary tradeoff in response to amino acid identity", in which they study phenotypic diversification of yeast cells upon exposure to specific sources of nitrogen, limited or replete.

Phenotypic diversification of isogenic yeast populations has emerged as an important consideration of yeast physiology and of evolutionary selection process as well as a useful model for understanding the phenotypic plasticity of cells in general. The topic of this manuscript is timely and important.

The authors present a large body of experimental work on a high technical level. The experiments they present are challenging, they are performed in a sound fashion and with all necessary controls. (In this regard it is to be especially highlighted the authors' demonstration of the observed effect being indeed phenotypic and non-genetic, somewhat unnecessarily hidden in the Supplementary Note 1).

However, some of the central conclusions that authors make seem unjustified (see below) and, unless upheld by further evidence, the manuscript consequently lacks clear novelty. In the current form, I cannot recommend the manuscript for publication.

Major comments:

1. The authors use the single cell RNA seq data from Jackson et al. to find a marker to distinguish the phenotypic subpopulation, and end up choosing Rpl28, which seems like a well-reasoned choice. Subsequently, in Fig. 2A, they find that this marker correlates remarkably well with cell size. From Supplementary Figure 3C and 3D they draw the conclusion, that Rpl28 distinguishes the two subpopulations better than merely the cell size. They do so based on the very low p-values in Fig. S3D; however, with large sample size, p-value does not reflect magnitude of the difference very well. To drive their point that Rpl28 is indeed a better marker than merely the cell size, authors should quantify AUROC for the Rpl28 expression normalised to cell size to classify cells into Rpl28high or Rpl28low. I would expect this AUROC to be pretty low, actually leading to the conclusion that cell size is basically as good a marker for the two phenotypic subpopulations as the best transcriptional marker. This does not undermine further conclusions of the authors and I actually find this to be a positive result; the authors draw the opposite conclusion though.
2. The tradeoff between growth rate, lag time after media replenishment and survival in starvation for the phenotypically diversified subpopulations has in principle been previously reported (e.g. in ref. 36). This is not novel, and should not be presented as such in the manuscript.
3. Related to the previous point, what indeed is novel, is the authors exploration of how the lag times and starvation survival rates depend on the identity of the nitrogen source (Fig. 4). This is a nice generalisation of the previous report of the existence of such trade-off, in the sense that this trade-off likely exists irrespective of how exactly the phenotypic diversity is induced. However, I have doubts about the quantitative characterisation of this trade-off that the authors attempt to give in Fig. 3C. In terms of growth rate, are the authors plotting the growth rate of the Rps28high subpopulation only or an average growth rate of the entire population? I understand it is the latter. But then Fig. 3C is a trivial result, as the higher the fraction of cells in the slow growing state, the lower the average growth rate of the population will be. What would indeed be interesting is if the extent of bimodality (or better still, fraction of cells in the slow growing subpopulation) would be related to the growth rate of the Rps28high population.
4. Related to previous point, assuming that the result in Fig. 3C is trivial, the differences between amino acids in their ability to induce phenotypic diversification remain wholly unexplained. This is especially puzzling given the authors' finding that it is the identity, not as much the quantity of the amino acid, what matters for determining the extent of phenotypic diversification.

Minor comments:

5. Related to the previous, the authors talk about the 'quality' of nitrogen source. They should either define the 'quality', or replace it with 'identity', 'type', or some other word not suggesting

there is a deeper understanding of the differences between these nitrogen sources in their ability to induce phenotypic diversification.

6. I do not see any experiment or analysis that would justify the 'evolutionary' word in the title. A trade-off selected by evolution is something else than an evolutionary trade-off.

7. I suggest that authors cite additional work that characterizes the subpopulations of yeast cells under glucose starvation, which share important features reported here by the authors (e.g. unbudded daughters being in the stress resistant population), so as to clarify the scope of novelty of their work. E.g. PMIDs: 21289090 and 16818721.

Reviewer #2 (Remarks to the Author):

Phenotypic heterogeneity in lifespan is influenced by an evolutionary tradeoff in response to amino acid identity.

This study by Shabestary et al. conducted on the laboratory yeast *S. cerevisiae* reveals that when faced with a nitrogen downshift, populations differentiate into two subpopulations focusing on growth or lifespan extension.

The metabolic response of these subpopulations is characterized using various techniques, showing differences in MAPK signaling and ATP content.

This research sheds light on the strategic choices organisms make between growth and survival under changing nutrient conditions.

This study is inspiring. It suggests as to how eukaryotes strategically respond to nutritional change.

Analyses of two subpopulations are technically novel and sound.

The manuscript was written in an excellent manner: description and explanation are reader-friendly, elegant and intelligent, unlike many other papers regarding something-omics.

Major points.

LL.160-165. The authors mentioned that the heterogeneity was "reversible".

This is indeed interesting. Then, what will happen a subpopulation is repeatedly exposed to another limited nitrogen environment? e.g., if the population G1 in the GLN medium is collected and then shifted to the PRO medium, do they immediately sense it and separate into P1 and P2 subpopulations? Alternatively, the G1 cells might stay in a single population even after shift to the PRO medium, which might indicate that the 'reset' requires a shift-back to the rich medium YPD.

Also, if the reversible heterogeneity exists, it could reflect the history or character of each individual cell. When two subpopulations was made by medium shift from YPD to PRO, each individual cell in P1 and P2 might record it on its own. When the two subpopulations were reunited as a single population by YPD, and then re-separated into two subpopulations P1' and P2' by PRO, are P1' cells previous P1 cells, and P2' cells = previous P2 cells? I am wondering if each cell has its own memory of previous starvation epigenetically or its own tendency regarding which behaviours individual cells prefer.

LL.170-173. "To investigate the dynamics of the differentiation between high and low subpopulations, we performed a time-course analysis for nitrogen-limited proline and glutamine conditions and monitored cell size and pRPL28 fluorescence every two hours for 8 h (Fig. 2a)"

I appreciate the elegant system using the promoter of RPL28, which may reflect transcriptional regulation upon nitrogen shift. But what was monitored there was pRPL28-driven protein levels of GFP as fluorescence. Is it possible that GFP downregulation as well as translational regulation are also included to the behaviour of pRPL28-driven GFP fluorescence? What is the half-life of GFP?

Fig. 5e,f

The authors nicely demonstrated that the high subpopulation has an advantage in growth, whereas the low subpopulation has an advantage in longevity. Ideally, this determination of cell fate would

be largely linked to transcriptional regulation, if not all, then the differentiated activity (nuclear localisation) of transcription factors could be a key for the determination. In Fig. 5f, Rdr1, Hap3... seem active in low-Pro, and Wtm1, Bcy1,... seem inactive in low-Pro (or active in high-Pro). I wonder if these TFs (Wtm1,,,) active in high-Pro indeed confer advantages in growth to the high subpopulation? Alternatively, wondering if the other TFs (Rdr1, Hap3,,,,) active in low-Pro might indeed confer resistance to stress. Deletion analyses would be feasible to see what would happen when such TF was deleted: whether two subpopulations are still made, or whether the advantage/disadvantage is disturbed by the deletion.

Minor points.

In Figure 1c, GS (growth scores) with waves are shown, which is not explained to a wide range of readers. It should be briefly explained for readers so that they do not have to read the cited literature.

L. 274 "Among those, ammonium, arginine, asparagine, methionine, and serine gave macro-homogenous profiles"

It may be better to add "downshift to" or "shift to" in front of "ammonium" just in case.

Fig. 6a,b LL442-449

It was very hard to understand how the ATP biosensor works and how to digest and interpret the data presented in Fig. 6a. Some graphical images would help readers' understanding.

Reviewer #3 (Remarks to the Author):

In this manuscript, Shabestary et al show the impact of nitrogen limitation on cellular metabolism at the single-cell level using the budding yeast as an experimental model. Wild type environments are never static and cells are all the time exposed to fluctuating nutrient conditions and they are forced to adapt, with lack of adaptation leading to population extinction. What would happen when the limiting ingredient is nitrogen? Is it the amount of the limitation or the type of the limited class that makes the downstream impact? These are excellent basic science questions this study went after.

More specifically, the authors used subpopulation markers (allowing interrogation of cellular fate at the onset of a nitrogen downshift) and studied differentiation across 24 different nitrogen sources with certain nitrogen sources being systematically limited in the media.

The authors report the presence of both isogenic quiescent and growing subpopulations; these showed differences in cell size, chronological lifespans and ability to resume growth. In other words, in response to nitrogen limitation, this study elucidated phenotypic heterogeneity composed of two subpopulations. How could this be useful for the cell populations going through the stress of nitrogen limitation? This heterogeneity could facilitate a population-level adaptation strategy as the two underlying isogenic subpopulations are metabolically specialised in either growth or survival.

The set of characterizations performed in the study build towards a novel amino acid dependent mechanism contributing to population structure and dynamics. Among the other interesting results or arguments based on the results of this study are cell size heterogeneity's potentially being a regulatory mechanism in response to nitrogen downshift and leucine's potentially being an environmental cue for dispersion. I think this study would make a large impact in multiple subdisciplines of basic biology.

I have one major and two minor comments/suggestions for the authors:

Major suggestion:

The method through which the authors quantified chronological lifespan (CLS) was not a field

standard method. The way CLS is measured involves waiting cells in a glucose restricted media and measure their ability to revive from a "paused" state. This difference between the restriction agents should be acknowledged in the manuscript. More importantly, though, in the Methods section describing the CLS assay, the authors write "Lifespan was measured up to 30 days after the end of the post-shift as indicated in text" and they use PI dye to distinguish between dead/live cells. In the standard CLS assay, it's the cells ability to regrow in 2% glucose at specific time points the cells are sampled from the glucose restricted media (not just whether or not they are dead/live at the time of sampling), therefore it is necessary that the authors should expose the cells to rich media throughout the 30 days time frame (at the chosen time points during the 30 days period).

Minor suggestions:

1. The terms "macro homogeneity" and "macro homogeneity" seems excessive when, in the literature, bimodality and monomodality have been used extensively (while there are numerous examples, see for example this paper in which both bimodality of both states and heterogeneity of the active state were quantified doi.org/10.1038/ncomms12959). Shabestary et al later make use of bimodality in their manuscript but perhaps they should be more consistent.

2. Most of the yeast aging studies uses replicative lifespan. To highlight the true nature of the authors' actual assay, the title should be modified to "Phenotypic heterogeneity in YEAST CHRONOLOGICAL lifespan follows an evolutionary tradeoff in response to amino acid identity."

REVIEWER COMMENTS

Reviewer #1 (Remarks to the Author):

Shabestary et al. present a manuscript titled "Phenotypic heterogeneity in lifespan follows an evolutionary tradeoff in response to amino acid identity", in which they study phenotypic diversification of yeast cells upon exposure to specific sources of nitrogen, limited or replete.

Phenotypic diversification of isogenic yeast populations has emerged as an important consideration of yeast physiology and of evolutionary selection process as well as a useful model for understanding the phenotypic plasticity of cells in general. The topic of this manuscript is timely and important.

The authors present a large body of experimental work on a high technical level. The experiments they present are challenging, they are performed in a sound fashion and with all necessary controls. (In this regard it is to be especially highlighted the authors' demonstration of the observed effect being indeed phenotypic and non-genetic, somewhat unnecessarily hidden in the Supplementary Note 1).

However, some of the central conclusions that authors make seem unjustified (see below) and, unless upheld by further evidence, the manuscript consequently lacks clear novelty. In the current form, I cannot recommend the manuscript for publication.

We thank the reviewer for his/her assessment of our manuscript and recognizing that the topic is timely and important. On the "lack of novelty" assessment, our results go beyond previous studies in multiple ways, summarised into three main points:

- 1) Establishment of amino acids as key regulators of cell size and population dynamics. As pointed out later by the reviewer, this is the first study that assesses the impact of a single nitrogen source on *S. cerevisiae* metabolism. Our results, both at the population and at the single-cell levels show that cell size resumption in *S. cerevisiae* is responsive to the identity of the nitrogen source, which was previously unknown. This resulting response has a population-wide effect with the formation of distinct subpopulations with synergetic specialisation. Given that the heterogeneity observed cannot be explained by our current understanding of nitrogen regulatory programs such as the nitrogen catabolite repression (NCR) or general amino acid control (GAAC), this study shows that nitrogen sensing and downstream decision making in *S. cerevisiae* is more complex than the current model. We also performed additional experiments using the biosensors to further assess metabolic differences across nitrogen sources.
- 2) Multi-level and dynamic analysis of the resulting quiescent and non-quiescent state. Our analysis further reveals some of the molecular basis that distinguish quiescent from non-quiescent fractions. We show that differentiation into a second fraction enables a wider metabolic specialisation that allows a population-wide adaptation. More importantly, by comparing sorted fractions with unsorted fractions we show that investment in a quiescent fraction does not lead to major fitness

decrease when re-exposed to nitrogen replete conditions, proving the bet-edging utility of such differentiation.

This point goes beyond past studies of quiescence. First, quiescence has been studied in isolation from its population fitness benefit, where quiescent cells are typically isolated from stationary phase cultures. Here, inducing the quiescent state by shifting cells to nitrogen-limited media allows us to follow the differentiation from entry into and exit from quiescence and report population dynamics and fitness measurement both in terms of chronological lifespan and growth resumption, which is, to the best of our knowledge, novel. Second, this study is the first example of combining multiple tools to allow tracking of the cellular differentiation into quiescence *in vivo* and over time. Past studies have performed transcriptome analysis (Klosinska et al., 2011) or proteome analysis (Davidson et al. 2011) of the already differentiated state without such a continued dynamic tracking.

As suggested by the reviewer later on, we have also clarified the novelty of our study in the text:

*“... With these fundamentally opposite metabolic states, individual cells need to commit to either state (specialisation) and phenotypic bimodality is therefore optimal given finite resource allocation. A similar phenotypic specialisation into a growing and a stress-resistant fraction has also been found when *S. cerevisiae* is exposed to growth inhibitors³⁸. In proline, the low subpopulation is a terminal state enriched with daughter cells, while in glutamine, it can resume growth. Asymmetric generation of quiescent-specific daughter cells has also been observed during glucose exhaustion, at the onset of stationary phase¹² ... ” (lines 494-501, p.15)*

- 3) Tool development to perform the above quantification. For this project, we have introduced and/or developed tools to study quiescence heterogeneity. We used metabolic sensors to provide an *in vivo* measurement of the differentiation. This is the first time that metabolic sensors have been employed to track such differentiation with interesting observations (heterogeneous ATP and stronger decrease of FBP as cells enter quiescence). This and the growth marker we identified based on recently available scRNAseq, will allow further assessment of the quiescent state at the single-cell level. In addition, we developed a prototrophic transcription factor library specifically for this project which could be useful to study nitrogen limitation in the future since the canonical GFP library is auxotrophic. This library can be also used more systematically, since the metabolism of auxotrophic libraries is influenced by amino acid supplementation (Mülleder *et al.* 2012, doi: 10.1038/nbt.2442).

Major comments:

1. The authors use the single cell RNA seq data from Jackson et al. to find a marker to distinguish the phenotypic subpopulation, and end up choosing Rpl28, which seems like a well-reasoned choice. Subsequently, in Fig. 2A, they find that this marker correlates remarkably well with cell size. From Supplementary Figure 3C and 3D they draw the conclusion, that Rpl28 distinguishes the two subpopulations better than merely the cell size.

They do so based on the very low p-values in Fig. S3D; however, with large sample size, p-value does not reflect magnitude of the difference very well. To drive their point that Rpl28 is indeed a better marker than merely the cell size, authors should quantify AUROC for the Rpl28 expression normalised to cell size to classify cells into Rpl28^{high} or Rpl28^{low}. I would expect this AUROC to be pretty low, actually leading to the conclusion that cell size is basically as good a marker for the two phenotypic subpopulations as the best transcriptional marker. This does not undermine further conclusions of the authors and I actually find this to be a positive result; the authors draw the opposite conclusion though.

We thank the reviewer for the suggestion. We agree with the reviewer and it was not our intention to suggest that pRPL28 is a better marker than cell size but that using both pRPL28 and cell size is a better subpopulation discriminator than cell size alone at the *early stage* of the differentiation. For instance, we can see that cell sizes for the low and high subpopulation peaks overlap until 2 h post-shift (Supplementary Figure 5) and this is also particularly obvious when sorting after an hour of down-shift (Figure S1). We agree with the reviewer that for timepoints taken later than 2 h, then cell size alone is sufficient to sort high and low fractions. This is what we did for clustering the subpopulations when using the metabolic sensors from 4 h onwards (Figure 6 and Supplementary Fig. 24a).

Throughout the text, following the comment of the reviewer we have now made a further effort to clarify that sorting by cell size was enough to separate subpopulations. For instance, some changes now read as follows:

"... sorting on both fluorescence and cell size allowed for a more precise separation at the early stage of differentiation (Supplementary Note 1)." (line 171, p.5)

"... Since cell size was an overall good discriminator of differentiation, we measured cell size macro-heterogeneity 2 h, 4 h, 6 h, and 8 h into the shift as a measure of quiescence heterogeneity. As seen previously for NLIM-PRO and NLIM-GLN, measuring cell size distribution 6 h into the shift was enough to assess the dynamics of the low subpopulation and capture differences between bimodal and unimodal conditions (Fig. 2a)." (line 277-282, p.8-9)

2. The tradeoff between growth rate, lag time after media replenishment and survival in starvation for the phenotypically diversified subpopulations has in principle been previously reported (e.g. in ref. 36). This is not novel, and should not be presented as such in the manuscript.

On the novelty aspect, we believe it is important to highlight that most studies investigating physiological tradeoffs have been performed at the population level, where cells are typically exposed to different conditions and their physiology recorded at the whole population level (for instance: Basan *et al.* 2020, Nature, doi: 10.1038/s41586-020-2505-4). One of the novelties of this study is the observation of such tradeoffs at the single-cell level. To our knowledge, only a few studies have done so due to the technical difficulty to track and observe such populations in isogenic cultures. As pointed out by the reviewer, Lukačičin *et al.* (2022, Nature, doi: 10.1038/s41586-022-04633-0) have observed an intron-mediated trigger between resistance to stress and growth. Moreno-Gamez *et al.* have previously

described a tradeoff between growth and antibiotic resistance (Moreno-Gamez *et al.* 2020, PNAS, doi: 10.1073/pnas.2003331117). While these studies also look at the single-cell/subpopulation level, the context and mechanism behind the emergence of phenotypic heterogeneity is quite different from our study and so is the metabolic specialisation they are reporting (growth reduction, antibiotic resistance vs nitrogen limitation).

As pointed out by the reviewer in the following point, the other novelty of this study is about describing the importance of the amino acid identity to balance such tradeoffs. We show that cells specialised in either growth and survival coexist and could be involved in a more elaborate multicellular structure given that we could detect amino acids that were different from the amino acids added to the media. In addition, we also show a bet-edging effect of the differentiation since investment in the low subpopulation does not have a strong fitness effect for some of the observed conditions. This particular aspect has not been shown in the studies cited above. We have adjusted the text to highlight this particular aspect of the novelty and also performed additional experiments using the ATP sensor across all nitrogen sources (Supplementary Figure 25).

We have now revised the text to make sure we do not overstate the novelty of our findings. In addition, in an effort to be scientifically accurate, we have now contrasted our observations in the discussion section when referring to metabolic tradeoff. This now reads: "A similar phenotypic specialisation into a growing and a stress-resistant fraction has also been found when *S. cerevisiae* is exposed to growth inhibitors (Lukačičin *et al.* 2022)."

3. Related to the previous point, what indeed is novel, is the authors exploration of how the lag times and starvation survival rates depend on the identity of the nitrogen source (Fig. 4). This is a nice generalisation of the previous report of the existence of such trade-off, in the sense that this trade-off likely exists irrespective of how exactly the phenotypic diversity is induced. However, I have doubts about the quantitative characterisation of this trade-off that the authors attempt to give in Fig. 3C. In terms of growth rate, are the authors plotting the growth rate of the Rps28high subpopulation only or an average growth rate of the entire population? I understand it is the latter. But then Fig. 3C is a trivial result, as the higher the fraction of cells in the slow growing state, the lower the average growth rate of the population will be. What would indeed be interesting is if the extent of bimodality (or better still, fraction of cells in the slow growing subpopulation) would be related to the growth rate of the Rps28high population.

We believe the reviewer is referring to Fig. 4c where we show the first of the two "tradeoffs" presented in this figure. Fig. 4c is important as a validation of the growth prediction we made from the scRNAseq data analysis and selecting pRPL28 as "growth marker". In hindsight, this may seem trivial, but this results is a necessary experimental validation of the computational prediction using scRNAseq and the subpopulation maker. We have now added a line to highlight this statement which now reads: "As expected based on the pRPL28 growth marker, nitrogen sources sustaining higher maximal growth rate had a unimodal distribution, while nitrogen sources with bimodal distributions had poor growth, suggesting a strong anti-correlation (Fig. 4c; Spearman coefficient $\rho=-0.80$ and -0.74 for NLIM and NREP, respectively) between cell size bimodality and maximal achievable growth rate, further validating the computational growth prediction from the scRNAseq data."

The main tradeoff reported in this study is the one shown in Fig. 4h, which is the tradeoff between chronological lifespan and growth presented at the subpopulation level. Fig. 4c is therefore a necessary intermediate step that brings us to the more interesting results, such as those in Fig. 4h. In Fig. 4c, we show that the bimodality score is anti-correlated with growth of the whole population level. We believe the reviewer's suggestion to look at how the extent of bimodality is related to the growth rate of the high subpopulation is already part of the data. In the bimodal (previously called "macro-heterogeneous" in the first draft) conditions (NLIM-PRO, NSTARVE, etc.), the low fraction is the quiescent population which is not (or very poorly) contributing to growth. As such, for those conditions, the plot also does represent the relation between the extent of bimodality and the growth rate of the high subpopulation.

4. Related to previous point, assuming that the result in Fig. 3C is trivial, the differences between amino acids in their ability to induce phenotypic diversification remain wholly unexplained. This is especially puzzling given the authors' finding that it is the identity, not as much the quantity of the amino acid, what matters for determining the extent of phenotypic diversification.

We agree with the reviewer that the observation that quality is more important than quantity is an interesting one. We hypothesise that sensing mechanisms must be critical to the emergence (or disappearance) of the low subpopulation. For instance, data from NLIM-GLN clearly shows that the low subpopulation can resume growth in this condition. To this stage, we believe that TOR or other key components of nitrogen metabolism play an important role in triggering the growth resumption of low subpopulation (this being leaving the quiescent state, resuming cell size increase and replicative division).

How yeasts regulate their metabolism in response to nutrient availability - and in particular in response to nitrogen identity has been an active research topic for the past decades (Gonzalez *et al.* 2017, doi: 10.15252/embj.201696010). While unravelling how this regulation works is outside the scope of this study (and would require more than one study), we believe that our results could be a novel method to investigate amino acid sensing. For instance, knocking-out putative amino acid sensing proteins and recording the resulting phenotypic heterogeneity could be a way to validate those.

In this study, we have highlighted the role of the MAPK signalling pathway during differentiation both at the subpopulation RNAseq and TF localisation levels. We believe that extending our search to other nitrogen sources than NLIM-PRO and NLIM-GLN can shed light on how each nitrogen source affects differentiation. This is an avenue that our lab is interested in investigating but currently outside the scope of this study.

Minor comments:

5. Related to the previous, the authors talk about the 'quality' of nitrogen source. They should either define the 'quality', or replace it with 'identity', 'type', or some other word not suggesting there is a deeper understanding of the differences between these nitrogen sources in their ability to induce phenotypic diversification.

We thank the reviewer for the suggestion and have now changed "quality" to "identity" to stay consistent throughout the manuscript.

6. I do not see any experiment or analysis that would justify the 'evolutionary' word in the title. A trade-off selected by evolution is something else than an evolutionary trade-off.

This referred to the observation that the phenotypic differentiation studied here was also conserved in 20 wild isolates (Supplementary Data 3). In an effort to simplify the message brought by this study, we have now removed this term from the title and main text.

7. I suggest that authors cite additional work that characterizes the subpopulations of yeast cells under glucose starvation, which share important features reported here by the authors (e.g. unbudded daughters being in the stress resistant population), so as to clarify the scope of novelty of their work. E.g. PMIDs: 21289090 and 16818721.

We thank the reviewer for the suggestion. We have now referenced these two studies in our introduction and discussion. In particular, we refer to Allen et al. (2006; PMIDs:16818721) when mentioning the asymmetric entry into quiescence and Davidson et al. (2010; PMIDs: 21289090) when mentioning the opposite signatures between quiescent and non-quiescent metabolism. We would like to point out that both studies are about quiescence entry during glucose exhaustion in the stationary phase which is relatively different from nitrogen limitation with excess glucose, which is the focus of our study.

Reviewer #2 (Remarks to the Author):

Phenotypic heterogeneity in lifespan is influenced by an evolutionary tradeoff in response to amino acid identity.

This study by Shabestary et al. conducted on the laboratory yeast *S. cerevisiae* reveals that when faced with a nitrogen downshift, populations differentiate into two subpopulations focusing on growth or lifespan extension.

The metabolic response of these subpopulations is characterized using various techniques, showing differences in MAPK signaling and ATP content.

This research sheds light on the strategic choices organisms make between growth and survival under changing nutrient conditions.

This study is inspiring. It suggests as to how eukaryotes strategically respond to nutritional change.

Analyses of two subpopulations are technically novel and sound.

The manuscript was written in an excellent manner: description and explanation are reader-friendly, elegant and intelligent, unlike many other papers regarding something-omics.

We thank the reviewer for a positive evaluation of our manuscript.

Major points.

LL.160-165. The authors mentioned that the heterogeneity was “reversible”.

This is indeed interesting. Then, what will happen a subpopulation is repeatedly exposed to another limited nitrogen environment? e.g., if the population G1 in the GLN medium is collected and then shifted to the PRO medium, do they immediately sense it and separate

into P1 and P2 subpopulations? Alternatively, the G1 cells might stay in a single population even after shift to the PRO medium, which might indicate that the 'reset' requires a shift-back to the rich medium YPD.

The question asked by the reviewer is whether GLN alone is a sufficient condition to reset the differentiation (bimodal to unimodal) or whether all amino acids should be present at replete conditions (YPD). Since sorting the G1 population is challenging due to poor separability (not many cells in G2 and for a very short time at the onset of differentiation; Fig. 2a), we performed an additional experiment where we shifted cells to 2h GLN, and re-exposed them to 6h PRO, as well as 2h PRO followed by 6h GLN. We got a contrasted picture where subsequent shifts to GLN and PRO did not lead to bimodality, while a shift to PRO and then GLN slightly reduced the bimodality. Our results indicate that GLN can rescue the differentiation to some extent while PRO did not lead to further differentiation. Since the washing step can be hard to perform on cells that have been exposed to nitrogen-limited conditions, we could perform the washing step only once and cannot explode at this stage residual amino acids.

Legend: Cell size distribution for cells exposed to 8h glutamine (gln_8h), 8h proline (pro_8h), 2h glutamine followed by 6h proline (gln2pro), or 2h proline followed by 6h glutamine (pro2gln).

Also, if the reversible heterogeneity exists, it could reflect the history or character of each individual cell. When two subpopulations was made by medium shift from YPD to PRO, each individual cell in P1 and P2 might record it on its own. When the two subpopulations were reunited as a single population by YPD, and then re-separated into two subpopulations P1' and P2' by PRO, are P1' cells previous P1 cells, and P2' cells = previous P2 cells? I am wondering if each cell has its own memory of previous starvation epigenetically or its own tendency regarding which behaviours individual cells prefer.

This is indeed an interesting point. When we sorted high and low proline sorted fractions, re-exposed them individually to YPD and down-shifted them again to proline we did not see any

differences in their subpopulation profiles (Fig. S2 in Supplementary Note 1). We thus decided against pursuing further experiments, since there was no obvious memory effect. However, there could be underlying transcriptional or epigenetic differences not captured by flow cytometry, which is something worth investigating in future work. A setup where we could quickly exchange media such as a microfluidics device would enable such types of experiments in the future.

LL.170-173. “To investigate the dynamics of the differentiation between high and low subpopulations, we performed a time-course analysis for nitrogen-limited proline and glutamine conditions and monitored cell size and pRPL28 fluorescence every two hours for 8 h (Fig. 2a)”

I appreciate the elegant system using the promoter of RPL28, which may reflect transcriptional regulation upon nitrogen shift. But what was monitored there was pRPL28-driven protein levels of GFP as fluorescence. Is it possible that GFP downregulation as well as translational regulation are also included to the behaviour of pRPL28-driven GFP fluorescence? What is the half-life of GFP?

We thank the reviewer for bringing up this point. Indeed, many factors in addition to promoter activity are likely to contribute to final sfGFP levels. SfGFP is quite stable during normal growth conditions with reported half-lives above 20 h across model organisms (for instance: 10.2144/000112479, 10.1093/protein/12.12.1035) and a relatively quick maturation time (~5min; doi: 10.1038/nbt.2281). However, to our knowledge, these parameters have not been measured during nitrogen-limited conditions where activation of stress responses could potentially interfere with sfGFP half-life (for instance through the action of proteases). We believe that the reduction of sfGFP protein levels in the low subpopulation is a concerted action of decreased RPL28 synthesis (given the lower amount of transcripts measured through scRNAseq) and sfGFP protein partitioning between mother and daughter cells after cell division.

To test this hypothesis and show that the pRPL28 marker reflects RPL28 levels and not sfGFP dynamics, we have now conducted additional experiments where we fused GFP to RPL28 and could observe the same subpopulation dynamics (4h of NLIM-PRO) indicating that RPL28 protein levels are also lower in the low subpopulation (Now Supplementary Fig. 3e; see below). This is interesting because this would also suggest either asymmetry in RPL28 inheritance between “high” mother and “low” daughter cells or active RPL28 degradation in the “low” daughter cells or more active replenishment of pRPL28 in “high” cells due to higher RPL28 synthesis.

This additional experiment of course does not rule out that the low subpopulation could also be experiencing a decrease in global transcription. In fact, analysis of TF intensity shows lower TF expression levels in the “low” proline subpopulation compared to the “high” subpopulation at later timepoints. Given the intricate relationship between growth, ribosome levels and transcriptional rate, it is difficult to deconvolute lower RPL28 expression from global transcriptional decrease. However, in the model from Airoidi *et al.* (doi:10.1371/journal.pcbi.1000257) used to compute growth scores in Fig. 1, we note that pRPL28 expression is particularly sensitive to growth status indicating that lower pRPL28 expression cannot be explained by global transcriptional decrease alone. We have now added this additional data and this now reads: “Additionally, exposing *Rpl28* tagged GFP

strains to 4 h of proline treatment could also show the emergence of a “low” subpopulation with lower Rpl28-GFP levels (Supplementary Fig. 3e), indicating that lower fluorescence in the “low” subpopulation was not due to post-transcriptional regulation of sfGFP alone.”

Supplementary Figure 3. (e) Signal comparison between pRPL28 driven sfGFP and Rpl28-tagged GFP fluorescence. Data is 4 h post-shift.

Fig. 5e,f

The authors nicely demonstrated that the high subpopulation has an advantage in growth, whereas the low subpopulation has an advantage in longevity. Ideally, this determination of cell fate would be largely linked to transcriptional regulation, if not all, then the differentiated activity (nuclear localisation) of transcription factors could be a key for the determination. In Fig. 5f, Rdr1, Hap3.... seem active in low-Pro, and Wtm1, Bcy1,... seem inactive in low-Pro (or active in high-Pro). I wonder if these TFs (Wtm1,,,) active in high-Pro indeed confer advantages in growth to the high subpopulation? Alternatively, wondering if the other TFs (Rdr1, Hap3,,,) active in low-Pro might indeed confer resistance to stress. Deletion analyses would be feasible to see what would happen when such TF was deleted: whether two subpopulations are still made, or whether the advantage/disadvantage is disturbed by the deletion.

We thank the reviewer for the suggestion. Indeed, looking at the ability of the knockouts to form subpopulations would be interesting and something we had considered. However, we think it would be difficult at this stage to select transcription factors that are actually upstream of the main control point(s) from the ones that are downstream. In addition, most of these transcription factors are essential or critical in regulating multiple aspects of cellular physiology (Bcy1 from the PKA pathway for instance). Therefore, deletion and analysis of the downstream response alone would not provide a sufficient mechanistic validation of the TF role in the differentiation in our opinion. Instead, we believe that in order to achieve this, more conditions should be tested (i.e. expanding our screen to all nitrogen conditions). This is what we are currently working on for a follow-up study.

Minor points.

In Figure 1c, GS (growth scores) with waves are shown, which is not explained to a wide range of readers. It should be briefly explained for readers so that they do not have to read the cited literature.

We thank the reviewer for highlighting this point. We have now added information to the caption to help the reader. This now reads: "Cells in the top UMAP plot are coloured by growth scores, calculated from a regression model trained on bulk RNAseq data. Histogram above represents the density of growth scores for each condition."

L. 274 "Among those, ammonium, arginine, asparagine, methionine, and serine gave macro-homogenous profiles"

It may be better to add "downshift to" or "shift to" in front of "ammonium" just in case.

We agree with the reviewer's advice and have corrected this.

Fig. 6a,b LL442-449

It was very hard to understand how the ATP biosensor works and how to digest and interpret the data presented in Fig. 6a. Some graphical images would help readers' understanding.

We have followed the reviewer's advice and added a schematic of how the sensors work as part of Fig. 6 (now Fig. 6a and 6b).

Reviewer #3 (Remarks to the Author):

In this manuscript, Shabestary et al show the impact of nitrogen limitation on cellular metabolism at the single-cell level using the budding yeast as an experimental model. Wild type environments are never static and cells are all the time exposed to fluctuating nutrient conditions and they are forced to adapt, with lack of adaptation leading to population extinction. What would happen when the limiting ingredient is nitrogen? Is it the amount of the limitation or the type of the limited class that makes the downstream impact? These are excellent basic science questions this study went after.

More specifically, the authors used subpopulation markers (allowing interrogation of cellular fate at the onset of a nitrogen downshift) and studied differentiation across 24 different nitrogen sources with certain nitrogen sources being systematically limited in the media.

The authors report the presence of both isogenic quiescent and growing subpopulations; these showed differences in cell size, chronological lifespans and ability to resume growth. In other words, in response to nitrogen limitation, this study elucidated phenotypic heterogeneity composed of two subpopulations. How could this be useful for the cell populations going through the stress of nitrogen limitation? This heterogeneity could facilitate a population-level adaptation strategy as the two underlying isogenic subpopulations are metabolically specialised in either growth or survival.

The set of characterizations performed in the study build towards a novel amino acid dependent mechanism contributing to population structure and dynamics. Among the other

interesting results or arguments based on the results of this study are cell size heterogeneity's potentially being a regulatory mechanism in response to nitrogen downshift and leucine's potentially being an environmental cue for dispersion. I think this study would make a large impact in multiple subdisciplines of basic biology.

We thank the reviewer for a positive assessment of our manuscript.

I have one major and two minor comments/suggestions for the authors:

Major suggestion:

The method through which the authors quantified chronological lifespan (CLS) was not a field standard method. The way CLS is measured involves waiting cells in a glucose restricted media and measure their ability to revive from a "paused" state. This difference between the restriction agents should be acknowledged in the manuscript. More importantly, though, in the Methods section describing the CLS assay, the authors write "Lifespan was measured up to 30 days after the end of the post-shift as indicated in text" and they use PI dye to distinguish between dead/live cells. In the standard CLS assay, it's the cells ability to regrow in 2% glucose at specific time points the cells are sampled from the glucose restricted media (not just whether or not they are dead/live at the time of sampling), therefore it is necessary that the authors should expose the cells to rich media throughout the 30 days time frame (at the chosen time points during the 30 days period).

We thank the reviewer for the suggestion and agree that CLS is often measured through re-exposure to caloric restricted or rich media (generally with complete amino acid supplement). However, this method is very specific to study CLS in glucose starved cultures and, in our case, not feasible when comparing survival between low and high fractions because re-exposing low and high fractions to complete amino acid supplement will abort and reinitialize the differentiation, preventing direct comparison between low and high fraction (Supplementary Note 1).

However, we agree that measuring viability is not the same as measuring actual survivability, as pointed out by the reviewer. In an effort to be more accurate scientifically, we are now referring to viability instead of survival and have now swapped these terms throughout the manuscript.

Minor suggestions:

1. The terms "macro homogeneity" and "macro homogeneity" seems excessive when, in the literature, bimodality and monomodality have been used extensively (while there are numerous examples, see for example this paper in which both bimodality of both states and heterogeneity of the active state were quantified doi.org/10.1038/ncomms12959). Shabestary et al later make use of bimodality in their manuscript but perhaps they should be more consistent.

We thank the reviewer for the suggestion and bringing this study to our attention. We have now re-branded the terms "macro homogeneity" and "macro heterogeneity" to unimodality

and bimodality, when relevant, to stay consistent with previous literature and referenced doi.org/10.1038/ncomms12959 when referring to population modality.

2. Most of the yeast aging studies uses replicative lifespan. To highlight the true nature of the authors' actual assay, the title should be modified to "Phenotypic heterogeneity in YEAST CHRONOLOGICAL lifespan follows an evolutionary tradeoff in response to amino acid identity."

We thank the reviewer for the note and agree that lifespan is mostly studied through the prism of replicative lifespan. Given the previous comment, we have now changed this term for the term "viability," which is better suited in the context of this study.

REVIEWERS' COMMENTS

Reviewer #1 (Remarks to the Author):

The authors have now improved the presentation and addressed my concerns. I support the publication of the manuscript in the current form.

Minor comments:

1. The authors have used the term 'bet-edging' at several places in the Rebuttal Letter and the updated manuscript. Please correct to 'bet-hedging'.
2. For the data in Supplementary Fig. 25, please also summarise them as a bar chart capturing the median ATP sensor output for the given amino acid.
3. ATP levels have been shown to be a function of growth rate. Thus, for the data in Supplementary Fig. 25, please also show a scatter plot of median ATP sensor output vs maximum growth rate for the given amino acid as reported in Fig. 4b, even if the relationship was not significant.

Reviewer #2 (Remarks to the Author):

The authors addressed all the points I had raised, and the revised manuscript no longer presents any concerns. Novelty, technically sound, and the work is supported by sufficient experiments in quality and quantity --- everything is fine.

Reviewer #3 (Remarks to the Author):

The authors have sufficiently addressed my comments and suggestions. They should consider correcting the typo in "bet-edging" throughout the manuscript by using the correct form: "bet-hedging". This is a beautiful story and I warmly recommend the publication of this work in Nature Communications.

REVIEWERS' COMMENTS

Reviewer #1 (Remarks to the Author):

The authors have now improved the presentation and addressed my concerns. I support the publication of the manuscript in the current form.

Minor comments:

1. The authors have used the term 'bet-edging' at several places in the Rebuttal Letter and the updated manuscript. Please correct to 'bet-hedging'.

Corrected.

2. For the data in Supplementary Fig. 25, please also summarise them as a barchart capturing the median ATP sensor output for the given amino acid.

We have now added a subpanel (subpanel b, shown below) within Supplementary Fig. 28 (Previously labelled Supp. Fig. 25) displaying the median ATP output for each amino acid.

3. ATP levels have been shown to be a function of growth rate. Thus, for the data in Supplementary Fig. 25, please also show a scatter plot of median ATP sensor output vs maximum growth rate for the given amino acid as reported in Fig. 4b, even if the relationship was not significant.

We have now added a subpanel (subpanel c, shown below) within Supplementary Fig. 28 displaying the median ATP output for each amino acid (4h measurement) versus maximal growth rate from Fig. 4b as suggested.

Supplementary Figure 28. ATP sensor output for all NLIM conditions. (b) Median population ATP sensor levels for each amino acid. **(c)** Median population ATP sensor levels (4h) versus maximal growth rate as displayed in Figure 4b. None represents the NSTARVE condition.

Reflecting these changes, this now reads: *“This difference in ATP heterogeneity between low and high subpopulations was maintained for the other bimodal nitrogen sources (Supplementary Fig. 28a). Similarly, conditions that sustained higher maximal growth rates (Fig. 4b) had generally lower median ATP output at the population level (Supplementary Fig. 28b and 28c).”* (line 479-483)

Reviewer #2 (Remarks to the Author):

The authors addressed all the points I had raised, and the revised manuscript no longer presents any concerns. Novelty, technically sound, and the work is supported by sufficient experiments in quality and quantity --- everything is fine.

Reviewer #3 (Remarks to the Author):

The authors have sufficiently addressed my comments and suggestions. They should consider correcting the typo in “bet-edging” throughout the manuscript by using the correct form: “bet-hedging”. This is a beautiful story and I warmly recommend the publication of this work in Nature Communications.

Corrected.